

# A radial variable for de Sitter two-point functions

**Manuel Loparco[1]⋆, Jiaxin Qiao[2]† and Zimo Sun[3]‡**

**1** Fields and Strings Laboratory, Institute of Physics,
École Polytechnique Fédéral de Lausanne (EPFL),
Route de la Sorge, CH-1015 Lausanne, Switzerland
**2** Laboratory for Theoretical Fundamental Physics, Institute of Physics,
École Polytechnique Fédérale de Lausanne (EPFL), CH-1015 Lausanne, Switzerland
**3** Princeton Gravity Initiative, Princeton University, Princeton, NJ 08544, USA

⋆ manuel.loparco@epfl.ch , † jiaxin.qiao@epfl.ch , ‡ zs8479@princeton.edu

## Abstract

We introduce a "radial" two-point invariant for quantum field theory in de Sitter (dS) analogous to the radial coordinate used in conformal field theory. We show that the two-point function of a free massive scalar in the Bunch-Davies vacuum has an exponentially convergent series expansion in this variable with positive coefficients only. Assuming a convergent Källén-Lehmann decomposition, this result is then generalized to the two-point function of any scalar operator non-perturbatively. A corollary of this result is that, starting from two-point functions on the sphere, an analytic continuation to an extended complex domain is admissible. dS two-point configurations live inside or on the boundary of this domain, and all the paths traced by analytic continuation between dS and the sphere or between dS and Euclidean Anti-de Sitter are also contained within this domain.

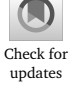

# 1  Introduction

In quantum field theories (QFTs) in flat space, under certain conditions, the Euclidean and Lorentzian correlation functions are related by analytic continuation with respect to the time variables [1–5]. Remarkably, similar results extend to QFTs defined on various other manifolds, including cylinders and anti-de Sitter spacetime. This success can be attributed to a common underlying factor: the presence of a conserved and positive Hamiltonian operator denoted as $H$, which generates (global) time translations via $e^{-iHt}$. The positivity of $H$ is a key factor: it ensures the well-definedness and analyticity of $e^{-iHt}$ within the complex domain $\mathrm{Im}(t) < 0$. Consequently, this leads to the analyticity of correlation functions in a certain complex domain of the time variables, including the Euclidean regime.

In the case of QFT in de Sitter (dS) and on the Euclidean sphere, instead, there is no conserved, positive and globally well-defined Hamiltonian.[1] Some alternative approach is thus required to prove the existence of an analytic continuation between the sphere and de Sitter. One approach was proposed by Bros, Epstein and Moschella in [6] (also mentioned in [7]), where they imposed the "weak spectral condition", namely assuming a minimal analyticity

---

[1]One may think about QFTs in the static patch of de Sitter, where there is a time-translation Killing vector. However, for the purpose of analytic continuation from the Euclidean sphere, the correlators are expectation values under a thermal state instead of a pure vacuum state. Then effectively $e^{-iHt}$ does not give exponential suppression of high-energy modes even when $\mathrm{Im}(t) < 0$.

domain of the correlation functions in complexified de Sitter (or equivalently, complexified sphere), inspired by the "spectral condition" in flat-space QFT. The "weak spectral condition" was proven to hold to all orders in perturbation theory by Hollands [8], but still lacks a non-perturbative proof. Using an argument similar to Bargman-Hall-Wightman theorem [9], the authors of [6] then extended the analyticity domain to what they called "maximal analyticity".

In this work, we are interested in proving that the "maximal analyticity" domain of two-point functions follows directly from assuming a convergent Källén-Lehmann decomposition into Unitary Irreducible Representations (UIRs) of the isometry group of de Sitter, $SO(d+1,1)$. In the process of deriving this result, we discover some facts which have interesting implications for how unitarity imposes constraints on de Sitter correlators. For other works concerning unitarity of dS correlators, see [6, 7, 10–24]. The analytic structure of dS correlators was also studied in [6, 7, 18, 25–32]. The importance of establishing the analyticity of correlators from de Sitter to the sphere is crucial for computations that are usually carried out in the Euclidean signature in order to avoid the IR divergences which plague de Sitter Feynman diagrams [27, 33–40].

The main results of this paper are better presented by introducing a new two-point invariant for de Sitter two-point configurations, which we call the *radial variable* (denoted as $\rho$). This variable is frequently employed in the context of the conformal bootstrap [41–44] and of the S-matrix bootstrap [45–47]. Then, all of our results are based on a key kinematical property of free propagators: the two-point function of a free scalar field in the principal or complementary series in the Bunch-Davies vacuum has a power series expansion in the radial variable $\rho$ which only includes positive coefficients. This fact is summarized in proposition 3.1 and is the first result of this paper.

This result implies the weak spectral condition of [6] in the following sense. Suppose we have a unitary two-point function on the Euclidean sphere (where $\rho \in [0,1)$ for the radial variable) which allows a convergent Källén-Lehmann representation in terms of free propagators of principal and complementary series. Then, as a consequence of proposition 3.1, the convergence domain of the Källén-Lehmann integral can always be extended from the real domain $\{\rho \in [0,1)\}$ to the complex domain $\{|\rho| < 1\}$. The latter is precisely the "maximal analyticity" domain of [6]. We have yet to establish whether all unitary scalar two-point functions on the Euclidean sphere satisfy our assumption of having a convergent Källén-Lehmann representation. However, we propose a practical way for verification, which is expected to be feasibly applicable in concrete examples.

We would like to emphasize that although each free propagator in the Källén-Lehmann representation is analytic in the domain $\{|\rho| < 1\}$, the analytic continuation of the whole two-point function is not obvious because one needs to swap the order of the Källén-Lehmann integral and the analytic continuation. The expansion of the two-point function in the radial variable makes it transparent: one can easily extend a convergent power series $\sum a_n \rho^n$ from $\rho \in [0,1)$ to $|\rho| < 1$, given that the expansion coefficients $a_n$ are all positive.

The existence and positivity of this radial expansion is thus an interesting new constraint that any QFT in dS has to satisfy, and it readily shows that the decomposition of the Hilbert space into UIRs implies the existence of an analyticity domain for two-point functions which connects two-point configurations in de Sitter, the sphere and Euclidean AdS.

**Outline**   This paper is organized as follows. In section 2, we review some preliminaries on dS spacetime, including the geometry and the scalar Unitary Irreducible Representations (UIRs) of its isometry group, $SO(d+1,1)$. In section 3, we introduce our basic assumptions and state our main results, i.e. proposition 3.1 and corollaries 3.2 and 3.3. Then, we discuss some implications of corollary 3.3, such as the analytic continuation of two-point functions from the Euclidean sphere to dS and to Euclidean anti-de Sitter space (EAdS). In section 3.3.3

we show an alternative simpler proof of corollary 3.3 which does not rely on the positive $\rho$ expansion. Section 4 is devoted to a proof of the proposition 3.1, which is technically the most challenging part of this work. In particular, the proof required the derivation of the Källén-Lehmann decomposition of principal and complementary free propagators in $dS_{d+1}$ in terms of free propagators of $dS_2$, which we derive in section 4.4. In section 5, we make conclusions and discuss some open questions and future directions.

## 2 Preliminaries

Before elaborating on our results, we will quickly review the geometry of dS and of the sphere, and the scalar UIRs of $SO(d+1,1)$. For the representation theory part, we mainly follow [48, 49].

### 2.1 Geometry

The $(d+1)$ dimensional dS spacetime can be embedded as a hyperboloid in $\mathbb{R}^{d+1,1}$

$$-(Y^0)^2 + (Y^1)^2 + \ldots + (Y^{d+1})^2 = R^2 \,, \tag{1}$$

where $Y^A \in \mathbb{R}^{d+1,1}$ and $R$ is the de Sitter radius. Scalar two-point functions in $dS_{d+1}$ depend on the $SO(d+1,1)$ invariant that can be constructed with two points

$$\sigma \equiv \frac{Y_1 \cdot Y_2}{R^2} \equiv \frac{1}{R^2} \eta_{AB} Y_1^A Y_2^B \,, \qquad \eta_{AB} \equiv \mathrm{diag}(-1,1,\ldots,1) \,. \tag{2}$$

Throughout this work, we will consider $Y_{1,2}^A$ as complex vectors in $\mathbb{C}^{d+1,1}$ satisfying eq. (1), and $\sigma$ as a complex variable. In particular, the regime with imaginary $Y^0$ and all other components real is the Euclidean sphere of radius $R$:

$$(\widetilde{Y}^0)^2 + (Y^1)^2 + \ldots + (Y^{d+1})^2 = R^2 \,, \tag{3}$$

where $\widetilde{Y}^0 = iY^0$. This is the Wick rotation from $dS_{d+1}$ to $S^{d+1}$. On the sphere, $\sigma$ becomes the Euclidean inner product between unit vectors in $\mathbb{R}^{d+2}$, taking values $-1 \le \sigma \le 1$. When the two points are antipodal to each other, $\sigma = -1$. When they are coincident, $\sigma = 1$. In de Sitter, instead, $\sigma$ can be any real number. When the two points are space-like separated, $\sigma < 1$. When they are time-like separated, $\sigma > 1$. Finally, light-like separation corresponds to $\sigma = 1$. See figure 1 for a representation of the complex $\sigma$ plane and where the physical configurations in each space lie. Without loss of generality, in the rest of this paper, we set $R = 1$.

### 2.2 Scalar fields in dS and UIRs of $SO(d+1,1)$

Given a free scalar field in $dS_{d+1}$ with a positive mass, its single-particle Hilbert space $\mathcal{H}$ carries a UIR of the isometry group $SO(d+1,1)$. Depending on the mass, $\mathcal{H}$ belongs to either the principal series or the complementary series [48–50]. It is convenient to parameterize the mass $m$ by a complex number $\Delta = \frac{d}{2} + i\lambda \in \mathbb{C}$ as follows

$$m^2 = \Delta(d-\Delta) = \frac{d^2}{4} + \lambda^2 \,. \tag{4}$$

The mass squared $m^2$ is positive when (i) $\lambda \in \mathbb{R}$ and (ii) $i\lambda \in \left(-\frac{d}{2}, \frac{d}{2}\right)$. In the first case, i.e. $m \ge \frac{d}{2}$, the single-particle Hilbert space $\mathcal{H}$ furnishes a principal series representation,

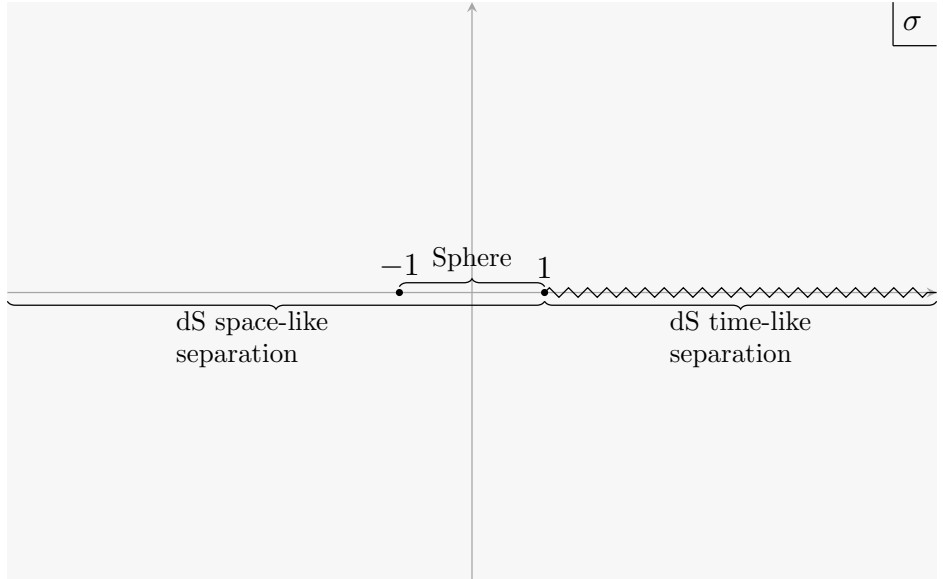

Figure 1: The physical values taken by $\sigma$ in dS and on the sphere. Free propagators in the Bunch-Davies vacuum have a branch cut at $\sigma \in [1, \infty)$.

denoted by $\mathcal{P}_\Delta$. In the second case, i.e. $m \in \left(0, \frac{d}{2}\right]$, $\mathcal{H}$ furnishes a complementary series representation, denoted by $\mathcal{C}_\Delta$. We sometimes use a uniform notation $\mathcal{F}_\Delta$ for both principal and complementary series, and its meaning is clear once we specify the value of $\Delta$. A detailed description of $\mathcal{F}_\Delta$ is reviewed in appendix E. We also want to mention that the mass is invariant under $\Delta \leftrightarrow \bar{\Delta} \equiv d - \Delta$. On the representation side, there is an isomorphism between $\mathcal{F}_\Delta$ and $\mathcal{F}_{\bar{\Delta}}$, established by the so-called shadow transformation [48, 51–54]. The isomorphism yields the "fundamental region" $\frac{d}{2} + i\mathbb{R}_{\geqslant 0} \cup \left(0, \frac{d}{2}\right)$ for $\Delta$.

The above massive representations cover most of the scalar UIRs of $SO(d + 1, 1)$. The remaining scalar UIRs are characterized by $m^2 = (1 - p)(d + p - 1)$ with $p$ being a positive integer. In these cases, the corresponding scalar field $\phi$ has a ($p$ dependent) shift symmetry. For example, when $p = 1$, $\phi$ is a massless scalar and hence the action is invariant under a constant shift. When $p = 2$, the shift symmetry becomes $\phi \rightarrow \phi + c_A Y^A$, where $c_A$ are constants. For higher $p$, the shift symmetry is described in detail in [55]. After gauging the shift symmetry, the single particle Hilbert space of $\phi$ carries the type $\mathcal{V}$ exceptional series $\mathcal{V}_{p,0}$ [49]. For $d = 1$, $\mathcal{V}_{p,0}$ is actually the direct sum of the highest and lowest weight discrete series $\mathcal{D}_p^\pm$, corresponding to the left and right movers along the global circle.

## 3 Main results and applications

### 3.1 Positive radial expansion in free theory

Our starting point is the Wightman two-point function of a massive free scalar with mass $m^2 = \frac{d^2}{4} + \lambda^2$ in $\text{dS}_{d+1}$. We choose to work in the Bunch-Davies vacuum, which is the unique dS invariant state that satisfies the Hadamard condition, and reduces to the correct Minkowski vacuum state in the flat space limit. Under proper normalization, the two-point function is given by

$$G_\lambda^{(d)}(\sigma) = \frac{\Gamma(\frac{d}{2} \pm i\lambda)}{(4\pi)^{\frac{d+1}{2}}} \mathbf{F}\left(\frac{d}{2} + i\lambda, \frac{d}{2} - i\lambda; \frac{d+1}{2}; \frac{1+\sigma}{2}\right), \tag{5}$$

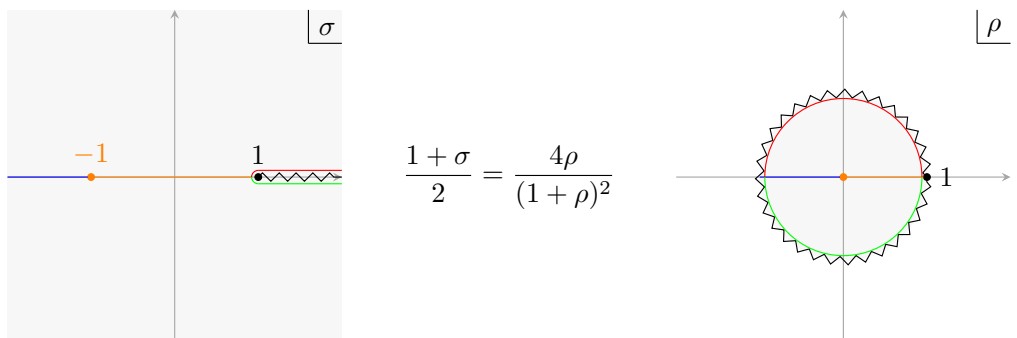

Figure 2: The analytic structure of the free propagators in the $\sigma$ and $\rho$ variables. The cut at time-like separation is mapped to a circumference in the $\rho$ complex plane. Antipodal separation, $\sigma = -1$, is mapped to the origin $\rho = 0$. The point corresponding to null separation is fixed in this transformation. The colors show how the physical values of $\sigma$ are mapped in the complex $\rho$ plane. The gray background shows that the full complex $\sigma$ plane is mapped to the unit $|\rho| < 1$ disc.

where we have used the shorthand notation $\Gamma(a \pm b) \equiv \Gamma(a+b)\Gamma(a-b)$, and by **F** we indicate the regularized hypergeometric function:

$$\mathbf{F}(a, b; c; z) = \frac{{}_2F_1(a, b; c; z)}{\Gamma(c)}. \tag{6}$$

Since we are considering a massive theory, i.e. $m^2 > 0$, the range of $\lambda$ is given by

$$\lambda \in \mathbb{R} \cup i\left(-\frac{d}{2}, \frac{d}{2}\right). \tag{7}$$

The hypergeometric function $\mathbf{F}(a, b; c; z)$ is known to be analytic on the cut plane

$$z \in \mathbb{C} \backslash [1, +\infty). \tag{8}$$

We introduce the **radial variable** $\rho$ which maps $z$ from the cut plane to the open unit disc reversibly:

$$\rho = \frac{1 - \sqrt{1-z}}{1 + \sqrt{1-z}}, \qquad z = \frac{4\rho}{(1+\rho)^2} \qquad (|\rho| < 1). \tag{9}$$

Under this change of variables, the free propagator $G_\lambda^{(d)}(\sigma(\rho))$ becomes an analytic function of $\rho$ on the unit disc, as shown in figure 2. For convenience, we will abuse the notation by writing the two-point function as $G_\lambda^{(d)}(\rho)$. The values taken by $\sigma$ on the sphere, $\sigma \in [-1, 1)$, are mapped to $\rho \in [0, 1)$. Space-like separated configurations in dS, corresponding to $\sigma \in (-\infty, 1)$, are mapped to $\rho \in (-1, 1)$. Time-like separation, corresponding to $\sigma \in (1, \infty)$ is mapped to the circumference $|\rho| = 1$ (but $\rho \neq \pm 1$) where, depending on the $i\epsilon$ prescription, we land close to the top half or the bottom half of the circumference. Infinite time-like or space-like configurations $\sigma = \infty$ are both mapped to $\rho = -1$.

The proposition that is at the basis of all our results can then be stated as follows:

**Proposition 3.1.** *The free propagators $G_\lambda^{(d)}(\sigma)$ for fields in the principal ($\lambda \in \mathbb{R}$) and the complementary ($i\lambda \in (-\frac{d}{2}, \frac{d}{2})$) series, have the expansion*

$$G_\lambda^{(d)}(\sigma) = \sum_{n=0}^{\infty} B_n(d, \lambda)\rho^n, \tag{10}$$

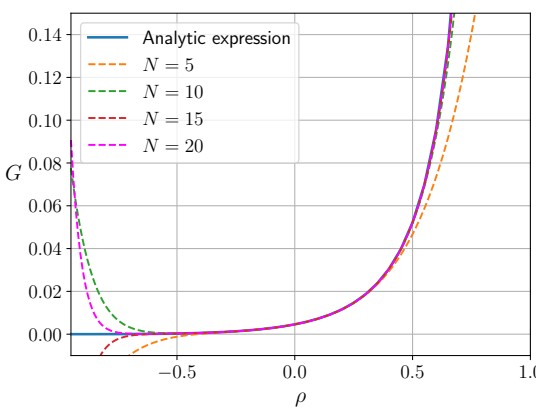

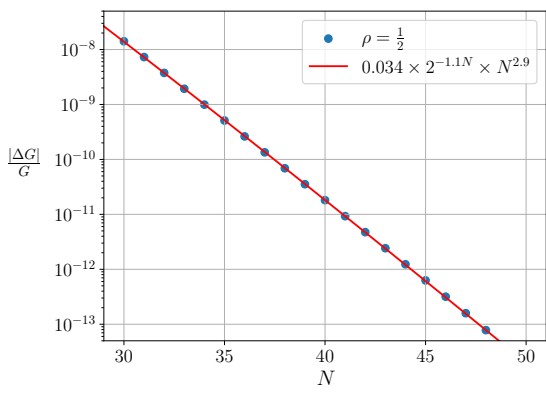

(a) The two-point function of a free scalar.

(b) Relative error as a function of $N$ at fixed $\rho$.

Figure 3: We compare the analytic expression of the two-point function of a free scalar (5) with $\lambda = 0.3$ in dS$_4$ with various truncations of the sum (10) as a function of the two-point invariant $\rho$. In subfigure 3a we plot the two-point functions themselves. In subfigure 3b we plot the relative error at fixed $\rho = \frac{1}{2}$ as a function of the truncation $N$. It decays exponentially.

with $B_n(d, \lambda) \geqslant 0$ for all n, and $\rho$ is defined implicitly through $\frac{1+\sigma}{2} = \frac{4\rho}{(1+\rho)^2}$.

The complete proof of this proposition is the most technical part of this work and we dedicate section 4 to it. For later convenience, we define

$$B_n(d, \lambda) = \frac{\Gamma(\frac{d}{2} \pm i\lambda)}{(4\pi)^{\frac{d+1}{2}} \Gamma(\frac{d+1}{2})} b_n(d, \lambda). \tag{11}$$

When $d = 3$, the first few $b_n$ are

$$b_0 = 1, \qquad b_1 = 2m^2, \qquad b_2 = \frac{4}{3}m^2(1 + m^2),$$

$$b_3 = \frac{2}{9}m^2(11 + 4m^2 + 2m^4),$$

$$b_4 = \frac{4}{45}m^2(30 + 22m^2 + 2m^4 + m^6), \tag{12}$$

where $m^2 = \frac{9}{4} + \lambda^2$, playing the role of mass as reviewed in section 2.2. These coefficients are all positive if and only if $m^2 > 0$, which holds when $\lambda \in \mathbb{R}$ or $i\lambda \in \left(-\frac{3}{2}, \frac{3}{2}\right)$. While we proved that $B_n(d, \lambda) > 0$ for all $n$ and $d$ for the complementary series, we did not manage to prove strict positivity for the principal series. Nevertheless, we expect that in fact $B_n(d, \lambda) > 0$ for all $n$ and $\lambda \in \mathbb{R} \cup i(-\frac{d}{2}, \frac{d}{2})$. The explicit expression of the coefficients $b_n(d, \lambda)$ is

$$b_n(d, \lambda) = \sum_{l=0}^{d} \sum_{k=0}^{n-l} \binom{d}{l} \frac{\left(\frac{d-1}{2}\right)_k (n-l-k)!}{k! \left(\frac{d+1}{2}\right)_{n-l-k}} \prod_{\pm} a_{n-l-k}\left(\frac{d+1}{4}, -\frac{d-1}{4} \pm i\lambda\right), \tag{13}$$

with $a_n$ defined in (55).

In figure 3 we show an example that the sum in (10) converges exponentially fast to the analytic expression (5) for fixed $\lambda$ and $d$, as the truncation $N$ increases. We will comment on the exact dependance of the relative error on $N$ in (20).

We do not have an intuitive or physical understanding of why the coefficients in (10) should be positive, beyond the very involved proof we present in section 4. Nevertheless, this nontrivial result leads to some interesting consequences as presented in the following subsections.

## 3.2 Positive radial expansion in interacting QFT

One direct application of proposition 3.1 is the fact that the series expansion in $\rho$ has to also be positive for all two-point functions in a general, interacting QFT.

To be precise, let us consider the Källén-Lehmann decomposition of the two-point function of a general scalar operator $\mathcal{O}$ in the Bunch-Davies vacuum [7, 17–19]:

$$G_{\mathcal{O}}(\sigma) \equiv \langle\Omega|\mathcal{O}(Y_1)\mathcal{O}(Y_2)|\Omega\rangle = \int_{\mathbb{R}} d\lambda \, \varrho_{\mathcal{O}}^{\mathcal{P}}(\lambda)G_{\lambda}^{(d)}(\sigma) + \int_{-\frac{d}{2}}^{\frac{d}{2}} d\lambda \, \varrho_{\mathcal{O}}^{\mathcal{C}}(\lambda)G_{i\lambda}^{(d)}(\sigma), \qquad (14)$$

where $\varrho_{\mathcal{O}}^{\mathcal{P}}(\lambda)$ and $\varrho_{\mathcal{O}}^{\mathcal{C}}(\lambda)$ are the spectral densities, supported on the principal and the complementary series respectively, $|\Omega\rangle$ is the interacting Bunch-Davies vacuum, and $G_{\lambda}^{(d)}(\sigma)$ is the two-point function (5).

Our basic assumptions are that

- Only free propagators of principal and complementary series with the choice of Bunch-Davies vacuum appear in the decomposition. We elaborate on the absence of exceptional type I and discrete series contributions in appendix A.

- The spectral densities $\varrho_{\mathcal{O}}^{\mathcal{P}}(\lambda)$ and $\varrho_{\mathcal{O}}^{\mathcal{C}}(\lambda)$ are positive.

- The integral is convergent for any pair of non-coincident points $(Y_1, Y_2)$ on the sphere.

It is worth noting that the above assumptions actually imply the two-point function is regular on the sphere (aside from coincident-point singularities), $SO(d+2)$ invariant and reflection positive. Despite the lack of proof, we conjecture that all two-point functions on the Euclidean sphere satisfying these Euclidean assumptions have a Källén-Lehmann decomposition of the form (14). We would like to also propose a way to check our assumptions in concrete examples, see the argument at the end of this subsection.

By the above assumptions and proposition 3.1, we can phrase the following corollary

**Corollary 3.2.** *Let $G_{\mathcal{O}}$ be a scalar two-point function on the Euclidean sphere $S^D$ ($D \geqslant 2$). If $G_{\mathcal{O}}$ has convergent Källén-Lehmann representation on the sphere of the form (14) with positive spectral densities, then $G_{\mathcal{O}}$ has a convergent power series expansion in the two-point invariant $\rho$ in the open unit disc $|\rho| < 1$, with nonnegative coefficients*

$$G_{\mathcal{O}}(\rho) = \sum_{n=0}^{\infty} c_{\mathcal{O},n}\rho^n, \qquad c_{\mathcal{O},n} \geq 0. \qquad (15)$$

The coefficient $c_{\mathcal{O},n}$ is given by the formula

$$c_{\mathcal{O},n} = \int_{\mathbb{R}} d\lambda \, \varrho_{\mathcal{O}}^{\mathcal{P}}(\lambda)B_n(d,\lambda) + \int_{-\frac{d}{2}}^{\frac{d}{2}} d\lambda \, \varrho_{\mathcal{O}}^{\mathcal{C}}(\lambda)B_n(d,i\lambda). \qquad (16)$$

The argument is as follows. Let us substitute (10) in the Källén-Lehmann decomposition (14):

$$G_{\mathcal{O}}(\rho) = \int_{\mathbb{R}} d\lambda \, \varrho_{\mathcal{O}}^{\mathcal{P}}(\lambda)\sum_{n=0}^{\infty} B_n(d,\lambda)\rho^n + \int_{-\frac{d}{2}}^{\frac{d}{2}} d\lambda \, \varrho_{\mathcal{O}}^{\mathcal{C}}(\lambda)\sum_{n=0}^{\infty} B_n(d,i\lambda)\rho^n, \qquad (17)$$

where we are abusing notation and using $G_{\mathcal{O}}(\rho)$ to indicate the same two-point function, highlighting its dependance on $\rho$, and $B_n(d,\lambda)$ was defined in (11). Because of our assumptions,

the spectral densities are positive and the integrals are convergent when $\rho \in [0,1)$. Then, according to proposition 3.1, the r.h.s. of (17) is absolutely convergent when $|\rho| < 1$, because

$$|G_{\mathcal{O}}(\rho)| \leq \int_{\mathbb{R}} d\lambda \, \varrho_{\mathcal{O}}^{\mathcal{P}}(\lambda) \sum_{n=0}^{\infty} B_n(d,\lambda)|\rho|^n + \int_{-\frac{d}{2}}^{\frac{d}{2}} d\lambda \, \varrho_{\mathcal{O}}^{\mathcal{C}}(\lambda) \sum_{n=0}^{\infty} B_n(d,i\lambda)|\rho|^n = G_{\mathcal{O}}(|\rho|). \quad (18)$$

Here, we have used the positivity of $B_n$. The absolute convergence allows us to swap the order of the sum and the integral. This justifies (15) with the coefficients given by (16). Let us make some remarks about this result

- In the previous subsection we have mentioned that $B_n(d,\lambda) > 0$ for the complementary series, while we can only state $B_n(d,\lambda) \geq 0$ for the principal series. Nevertheless, all numerical checks suggest that $B_n(d,\lambda) > 0$ for $\lambda \in \mathbb{R}$ too. A strong evidence is that for principal series, these coefficients can be expressed as a sum of squares (see (64)). This would immediately imply strict positivity for the $c_{\mathcal{O},n}$ coefficients too. Despite the lack of proof, we expect that generically $c_{\mathcal{O},n} > 0$. for all $n$.

- Since now $G_{\mathcal{O}}(\rho)$ is analytic on the open unit disc, the series expansion (15) converges exponentially fast for any fixed $\rho$:

$$\left| \sum_{n=N}^{\infty} c_{\mathcal{O},n} \rho^n \right| \leq G_{\mathcal{O}}(\rho_*) \left| \frac{\rho}{\rho_*} \right|^N \qquad (|\rho| < \rho_* < 1). \quad (19)$$

  If we further assume that the two-point function has power-law behavior at short distances: $G_{\mathcal{O}}(\rho) \sim const(1-\rho)^{-\alpha}$ as $\rho \to 1$, then the error term has the following bound:

$$\left| \sum_{n=N}^{\infty} c_{\mathcal{O},n} \rho^n \right| \lesssim const \, N^\alpha \, |\rho|^N. \quad (20)$$

  This estimate holds for sufficiently large $N$.[2]

Let us study an explicit example. Consider a unitary CFT in the bulk of dS spacetime. The (unit normalized) two-point function of a scalar primary operator with scaling dimension $\Delta$ is given by

$$G_{\text{CFT}}(\rho) = \frac{1}{2^\Delta(1-\sigma)^\Delta} = 2^{-2\Delta}\left(\frac{1+\rho}{1-\rho}\right)^{2\Delta}. \quad (21)$$

Here $\Delta \geq \frac{d-1}{2}$ as a consequence of unitarity. We can compute the associated $c_{\mathcal{O},n}$ coefficients with (16), using the spectral density computed in [17,19], or simply by Taylor expanding (21). The coefficients are positive.[3] In figure 4, we plot the two-point function reconstructed from the sum in (15) truncated at various values of $N$, and the relative error with respect to the analytic expression (21). We see that the relative error decreases exponentially as we increase $N$. Moreover, the fit we get for the relative error at large $N$ is consistent with the bound (20).

---

[2]Assuming the power-law behavior of the two-point function, when $N$ is sufficiently large, the minimum of the r.h.s. of (19) is around $\rho_* = \frac{N}{N+\alpha}$. Then substituting this value into (19) leads to (20).

[3]We have

$$\log G_{\text{CFT}}(\rho) = -2\Delta \log 2 + 4\Delta \sum_{k=0}^{\infty} \frac{\rho^{2k+1}}{2k+1}. \quad (22)$$

Then by exponentiating this expression we get a power series of $G_{\text{CFT}}(\rho)$ with positive coefficients.

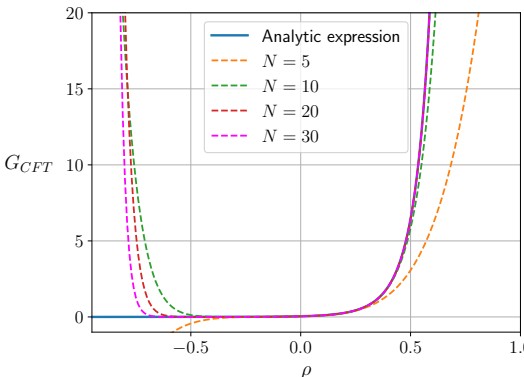

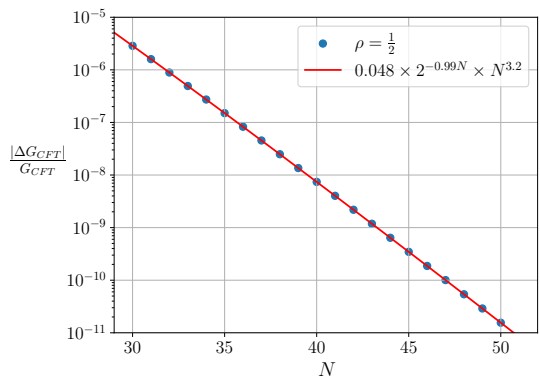

(a) The two-point function of a CFT scalar primary.

(b) The relative error as a function of $N$ at fixed $\rho$.

Figure 4: We compare the analytic expression of a two-point function of a scalar CFT primary operator in $\mathrm{dS}_4$ with scaling dimension $\Delta = 2.3$ (21) with what is obtained when truncating the sum (15) at various values of $N$, as a function of the two-point invariant $\rho$. In subfigure 4a we directly compare the two-point functions. In subfigure 4b we show that the relative error decays exponentially as we increase the truncation at fixed $\rho = \frac{1}{2}$.

Before concluding this subsection, we would like to comment on how to verify (14) in concrete examples. Suppose we are given a two-point function $G(X,Y)$ on the Euclidean sphere, and we expand it in terms of spherical harmonics [33]:

$$G(X,Y) = \sum_{\ell=0}^{\infty} \sum_m \frac{Y_{\ell,m}(X)Y_{\ell,m}^*(Y)}{\ell(\ell+d)+B(\ell)}, \tag{23}$$

where $\ell$ and $m = (m_1, \ldots, m_d)$ are the indices for the spin-$\ell$ representation of $SO(d+2)$, and $Y_{\ell,m}$ are the spherical harmonics on $S^{d+1}$. The explicit form of $Y_{\ell,m}$ is not crucial for our discussion. What is useful is that when $B(\ell) = M^2$, the above expansion yields the free propagator (5) with mass $M^2 = \frac{d^2}{4} + \lambda^2$. Therefore, determining the spectral density $\rho(M^2)$ in the Källén–Lehmann decomposition amounts to solving the following integral equation:

$$\frac{1}{\ell(\ell+d)+B(\ell)} = \int_0^{\infty} dM^2 \, \rho(M^2) \frac{1}{\ell(\ell+d)+M^2}. \tag{24}$$

Let us denote $x \equiv \ell(\ell+d)$, $f(x) \equiv \frac{1}{\ell(\ell+d)+B(\ell)}$, and $y \equiv M^2$. In this notation, the above equation becomes:

$$f(x) = \int_0^{\infty} dy \frac{\rho(y)}{x+y} = \int_0^{\infty} dy \int_0^{\infty} ds \, \rho(y)e^{-s(x+y)} = \mathcal{L}^2[\rho](x), \tag{25}$$

where $\mathcal{L}[\rho]$ denotes the Laplace transform of $\rho$. Thus, to obtain the spectral density $\rho$, one needs to perform the inverse Laplace transform twice on $f(x)$. Since $f(x)$ was originally defined only for discrete values of $x$, to perform the inverse Laplace transform, $f(x)$ needs to be extended to an analytic function on the domain $\mathrm{Re}(x) > 0$, similar to how partial waves are extended to complex angular momenta in Regge theory [56]. Assuming these operations can be performed, one can check (a) the positivity of $\rho$ and (b) the growth of $\rho$ to ensure the convergence of the Källén–Lehmann decomposition. We leave these considerations for future study.

## 3.3 Analyticity of the two-point function

By the argument presented in section 3.2, the two-point function $G_{\mathcal{O}}(\rho)$ is shown to be analytic within the open unit disc ($|\rho| < 1$). This domain of $\rho$ corresponds to $\sigma \in \mathbb{C}\backslash[1, +\infty)$ through an analytic mapping. Consequently, we conclude the following corollary:

> **Corollary 3.3.** *Let $G_{\mathcal{O}}(\sigma)$ be a two-point function with a convergent Källén–Lehmann representation of the form (14) on the interval $\sigma \in [-1, 1)$, corresponding to two-point configurations on $S^{d+1}$. Additionally, assume that the spectral densities $\varrho_{\mathcal{O}}^{\mathcal{P}}(\lambda)$ and $\varrho_{\mathcal{O}}^{\mathcal{C}}(\lambda)$ are non-negative. Then, $G_{\mathcal{O}}(\sigma)$ has analytic continuation to the complex domain $\sigma \in \mathbb{C}\backslash[1, \infty)$.*

Let us make some remarks about this corollary

- The domain of $\sigma$ indicated in corollary 3.3 is referred to as the "maximal analyticity" domain in [7] (see proposition 2.2 there). It is essential to note that the starting point in [7] differs from that in this paper. There, the fundamental assumption is the analyticity of the two-point function within the "forward tube" domain (see eq. (28) for a specific definition of the forward tube). The maximal analyticity is then obtained by applying complex de Sitter group elements to the forward tube. In contrast, our paper starts with the convergence of the Källén-Lehmann representation on the Euclidean sphere, from which we derive the same domain of analyticity using the expansion in the $\rho$ variable.

- The domain of analyticity encompassed in this corollary includes all paths taken when performing analytic continuation from the sphere to de Sitter and from de Sitter to Euclidean Anti-de Sitter. Further elaboration on these points can be found in sections 3.3.1 and 3.3.2.

- In fact, assuming the convergence of the Källén-Lehmann representation on the Euclidean sphere, the analyticity property of the two-point function, as stated in proposition 3.3, can be derived in a simpler manner. Additional details on this aspect can be found in section 3.3.3. Consequently, the primary nontrivial contributions of this work are the positivity of the $\rho$-expansion coefficients, as demonstrated in proposition 3.1 and corollary 3.2.

### 3.3.1 From the sphere to dS

In this subsection, our aim is to demonstrate that all the paths taken during the Wick rotation from the sphere to de Sitter are entirely contained within the maximal analyticity domain of corollary 3.3, except for the end points of these paths.

As reviewed in section 2, both the sphere $S^{d+1}$ and the de Sitter spacetime $\mathrm{dS}_{d+1}$ can be considered as distinct submanifolds of the same complex hyperboloid, defined by the condition:

$$(Y)^2 \equiv -(Y^0)^2 + (Y^1)^2 + \ldots + (Y^{d+1})^2 = 1, \qquad Y^A \in \mathbb{C}. \tag{26}$$

Specifically, $S^{d+1}$ corresponds to the submanifold where $Y^0$ is imaginary, while all other components are real. On the other hand, $\mathrm{dS}_{d+1}$ corresponds to the submanifold where all components are real.

The analyticity of the two-point function $G_{\mathcal{O}}(Y_1, Y_2)$ as a function of $\sigma = Y_1 \cdot Y_2$ is established by corollary 3.3, and it holds within the complex domain defined as:

$$\sigma \in \mathbb{C}\backslash[1, +\infty]. \tag{27}$$

It is important to note that for any two-point configuration $(Y_1, Y_2)$ within the "forward tube" domain, defined by (28), the range of $\sigma$ is within (27) [8]. The forward tube is characterized by conditions on $Y_1$ and $Y_2$ as follows:

$$(Y_1)^2 = (Y_2)^2 = 1, \qquad \mathrm{Im}(Y_{21}) \in V_+, \tag{28}$$

where $Y_{ij} \equiv Y_i - Y_j$ for convenience, and $V_+$ represents the forward lightcone, defined as:

$$V_+ := \left\{ Y \in \mathbb{R}^{1,d+1} \;\middle|\; Y^0 > \sqrt{\sum_{a=1}^{d+1} (Y^a)^2} \right\}. \tag{29}$$

To illustrate this point further, let us calculate $\sigma$ for $(Y_1, Y_2)$ satisfying (28). First, we compute the inner product $Y_1 \cdot Y_2$, which simplifies to:

$$\sigma \equiv Y_1 \cdot Y_2 = 1 - \frac{(Y_{12})^2}{2} = 1 - \frac{1}{2}\left[ \mathrm{Re}(Y_{12})^2 - \mathrm{Im}(Y_{12})^2 + 2i\,\mathrm{Re}(Y_{12}) \cdot \mathrm{Im}(Y_{12}) \right]. \tag{30}$$

We observe that (28) implies that $\mathrm{Im}(Y_{12})$ is time-like, resulting in $\sigma$ being real only when $\mathrm{Re}(Y_{12})$ corresponds to space-like separation or is zero. However, in this case, $\sigma \leqslant 1 - \frac{1}{2}\mathrm{Re}(Y_{12})^2 + \frac{1}{2}\mathrm{Im}(Y_{12})^2 < 1$ because $\mathrm{Re}(Y_{12})^2 \geqslant 0$ and $\mathrm{Im}(Y_{12})^2 < 0$. This establishes that (28) implies (27).

To emphasize, domain (28) includes all two-point configurations on the sphere where $iY_{12}^0 > 0$, indicating that $Y_1$ is closer to the north pole than $Y_2$. The de Sitter two-point configurations are not contained within domain (28), but rather on its boundary. Similar to QFT in flat space, we anticipate that the de Sitter Wightman two-point function can be obtained by taking the limit of $G(Y_1, Y_2)$, as an analytic function, from domain (28). Depending on the causal relation between $Y_1$ and $Y_2$, there are two cases for the de Sitter two-point configurations:

- Space-like separation: This case corresponds to $\sigma < 1$. Although these de Sitter configurations are not included in domain (28), they fall within the broader analyticity domain (27). Therefore, the de Sitter Wightman two-point function is analytic at space-like two-point configurations.

- Time-like or light-like separation: This case corresponds to $\sigma \geqslant 1$ ($\sigma = 1$ for light-like separation). Two-point configurations within this regime are on the boundary of domain (27) or (28). In terms of functions, the two-point function is singular when $Y_1$ and $Y_2$ are light-like separated. When $Y_1$ and $Y_2$ are time-like separated, without additional input (e.g., conformal invariance), it is unclear whether the two-point function is analytic or not.

Therefore, assuming the convergence of the Källén-Lehmann representation (14) on the Euclidean sphere, we can perform the analytic continuation of the two-point function from the Euclidean sphere to de Sitter in the standard manner.

Here, we present a two-step algorithm for performing the analytic continuation in global coordinates:

**Step 1**. Analytically continue the two-point function $G_{\mathcal{O}}(Y_1, Y_2)$ to the domain characterized by the following conditions:

$$
\begin{aligned}
&Y_k^0 = \sinh(T_k), \qquad Y_k^a = \Omega_k^a \cosh(T_k) \qquad (k = 1, 2), \\
&-\frac{\pi}{2} < \mathrm{Im}(T_1) < 0 < \mathrm{Im}(T_2) < \frac{\pi}{2}, \quad \mathrm{Re}(T_k) \in \mathbb{R}, \quad \Omega_k^a \in S^d.
\end{aligned}
\tag{31}
$$

In this step, we can demonstrate that domain (31) is included in domain (28),[4] ensuring the analyticity of the two-point function.

**Step 2**. Let $T_k = t_k + i\theta_k$ and take the limit as $\theta_1$ and $\theta_2$ tend to zero:

$$G_{\mathcal{O}}(t_1, \Omega_1; t_2, \Omega_2) = \lim_{\theta_1, \theta_2 \to 0} G_{\mathcal{O}}(t_1 + i\theta_1, \Omega_1; t_2 + i\theta_2, \Omega_2), \tag{33}$$

from the domain (31). Here in the l.h.s., we use the notation $Y_k^0 = \sinh(t_k)$ and $Y_k^a = \Omega_k^a \cosh(t_k)$ for $a = 1, 2, \ldots, d+1$. The limit (33) exists as a function when $Y_1$ and $Y_2$ are space-like separated, as discussed above.

The existence of the limit (33) for a general real $(Y_1, Y_2)$ pair requires additional assumptions concerning the behavior of the two-point function at short distances $((Y_1 - Y_2)^2 \to 0)$. We make a natural assumption that on the Euclidean sphere, the two-point function has, at most, a power-law divergence at short distances. This assumption is formally expressed as follows:

$$G_{\mathcal{O}}(Y_1, Y_2) \leqslant \frac{A}{(1 - Y_1 \cdot Y_2)^\alpha} \qquad (-1 \leqslant Y_1 \cdot Y_2 < 1), \tag{34}$$

where $A$ and $\alpha$ are finite, positive constants that may depend on the specific model.

Due to assumption (34) and the positivity of spectral densities in the Källén–Lehmann representation (14), the two-point function is bounded from above as follows:

$$G_{\mathcal{O}}(Y_1, Y_2) \leqslant \frac{A'}{|\mathrm{Im}(T_1)\mathrm{Im}(T_2)|^{2\alpha}}, \tag{35}$$

where $T_1$ and $T_2$ are the same as the ones in (31), and $A' = A\left(\frac{\pi^2}{8}\right)^\alpha$.

Now, $G_{\mathcal{O}}(Y_1, Y_2)$ is analytic in complex $T_k$ and continuous in real $\Omega_k$ on domain (31), with the power-law bound (35). According to Vladimirov's theorem [57], it follows that the limit (33) exists in the sense of tempered distributions in $T_k$.

### 3.3.2 From dS to EAdS

The analytic continuation from de Sitter to Euclidean Anti-de Sitter [28–31] is implemented using planar coordinates defined as follows:

$$Y^0 = \frac{\eta^2 - \mathbf{y}^2 - 1}{2\eta}, \qquad Y^i = -\frac{y^i}{\eta}, \qquad Y^{d+1} = \frac{\eta^2 - \mathbf{y}^2 + 1}{2\eta}. \tag{36}$$

Here, $\eta$ ranges from $-\infty$ to $0$, and $\mathbf{y}$ belongs to $\mathbb{R}^d$.

Under these coordinates, the de Sitter metric is expressed as:

$$ds^2 = \frac{-d\eta^2 + d\mathbf{y}^2}{\eta^2}. \tag{37}$$

If we take $\eta$ to be imaginary, i.e., $\eta = \pm iz$, the metric above transforms into:

$$ds^2 = -\frac{dz^2 + d\mathbf{y}^2}{z^2}. \tag{38}$$

This metric represents Euclidean Anti-de Sitter (EAdS) with a radius equal to one, up to an overall minus sign.

---

[4]Let $T_k = t_k + i\theta_k$ in (31). By explicit computation, we have

$$\mathrm{Im}(Y_k^0) = \cosh(t_k)\sin(\theta_k), \qquad \mathrm{Im}(Y_k^a) = \Omega_k^a \sinh(t_k)\sin(\theta_k). \tag{32}$$

Therefore, in domain (31) we have $-\mathrm{Im}(Y_1), \mathrm{Im}(Y_2) \in V_+$, which implies $\mathrm{Im}(Y_{21}) \in V_+$.

Now, let us consider the two-point function, denoted as $G_{\mathcal{O}}(\eta_1, \mathbf{y}_1; \eta_2, \mathbf{y}_2)$, in the domain of complex $\eta_k$ given by:

$$\text{Re}(\eta_1), \text{Re}(\eta_2) \in (-\infty, 0), \qquad \text{Im}(\eta_1) \in (-\infty, 0), \qquad \text{Im}(\eta_2) \in (0, \infty). \tag{39}$$

It is important to note that within this domain, by definition, the following conditions hold:

$$-\text{Im}Y_1^0 > \sqrt{\sum_{a=1}^{d+1} \text{Im}(Y_1^a)^2}, \qquad \text{Im}Y_2^0 > \sqrt{\sum_{a=1}^{d+1} \text{Im}(Y_2^a)^2}, \tag{40}$$

so we get $\text{Im}(Y_{21}) \in V_+$. As argued in the previous subsection, the configuration satisfying $\text{Im}(Y_{21}) \in V_+$ falls within the domain of analyticity established by corollary 3.3. Consequently, we can confidently state that the two-point functions are analytic in terms of $\eta_k$ and continuous in $\mathbf{y}_k$ within the domain (39).

In particular, the two-point function is analytic around the regime where both $\eta_1$ and $\eta_2$ are purely imaginary (with the constraints in (39)). As mentioned at the beginning of this subsection, the two-point configurations in this regime are interpreted as EAdS configurations. However, there is a subtlety in that the two points are not situated within the same EAdS branch. To illustrate this, let us consider the range of $\eta_1$ and $\eta_2$ as given in (39). When $\eta_1$ and $\eta_2$ take on imaginary values, i.e., $\eta_1 = iz_1$ and $\eta_2 = iz_2$, we find that:

$$z_1 > 0, \qquad z_2 < 0. \tag{41}$$

Furthermore, we observe that under the planar coordinates as defined in (36), all the components become imaginary. To simplify the notation, let us denote $Y^A \equiv -iX^A$. Upon this substitution, it is straightforward to verify that:

$$X_1^2 = X_2^2 = -1, \qquad X_1^0 > 0, \qquad X_2^0 < 0. \tag{42}$$

This means that $X_1$ and $X_2$ are situated in two distinct branches of EAdS within the embedding space: $X_1$ belongs to the upper branch, while $X_2$ belongs to the lower branch. See figure 5 for a visual representation. This explains why the corresponding range of $\sigma = -X_1 \cdot X_2$ is $(-\infty, -1]$, signifying that the two points have a minimal distance $((X_1 - X_2)^2 \leqslant -4)$ since they are located on time-like separated EAdS surfaces.

Now, let us reach the Wightman two-point function in dS from EAdS. According to the analyticity domain (39), we take the following limit:

$$G_{\mathcal{O}}(\eta_1, \mathbf{y}_1; \eta_2, \mathbf{y}_2) = \lim_{z_1, z_2 \to 0^+} G_{\mathcal{O}}(\eta_1 - iz_1, \mathbf{y}_1; \eta_2 + iz_2, \mathbf{y}_2). \tag{43}$$

The existence of this limit can be justified using arguments similar to those presented in the previous subsection. Utilizing the assumed bound (34), we can demonstrate that the two-point function in the planar coordinates satisfies the following power-law bound for complex $\eta_k$ in domain (39):

$$|G_{\mathcal{O}}(\eta_1, \mathbf{y}_1; \eta_2, \mathbf{y}_2)| \leqslant \frac{A}{\left[\left(1 - \frac{\text{Re}(\eta_1^2)}{|\eta_1|^2}\right)\left(1 - \frac{\text{Re}(\eta_2^2)}{|\eta_2|^2}\right)\right]^{\alpha/2}} = A \left|\frac{\eta_1 \eta_2}{2\,\text{Im}(\eta_1)\,\text{Im}(\eta_2)}\right|^\alpha. \tag{44}$$

This bound ensures that the limit exists in the sense of tempered distributions in terms of $\eta_k$.

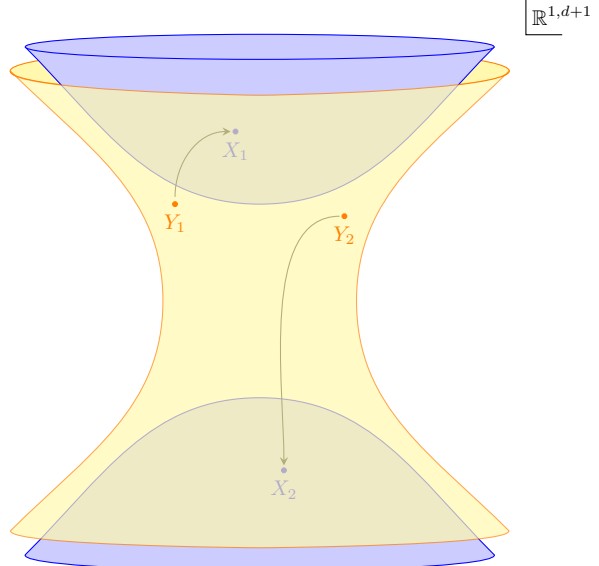

Figure 5: The analytic continuation of the Wightman two-point function $\langle\Omega|\mathcal{O}(Y_1)\mathcal{O}(Y_2)|\Omega\rangle$ from de Sitter to EAdS in embedding space. To avoid the cut at time-like separation, the two points are continued to two separate branches of EAdS.

### 3.3.3 A simple alternative derivation of analyticity

In this section we would like to present an alternative proof of corollary 3.3, which does not require the use of proposition 3.1.

The proof is divided into two steps. In the first step, we show analyticity of the two-point function in the "forward tube" domain (28). In the second step, we show that for $(Y_1, Y_2)$ in domain (28), the range of $\sigma \equiv Y_1 \cdot Y_2$ covers the whole cut plane, $\mathbb{C}\backslash[1, +\infty)$.

For the first step, the key observation is that each free propagator satisfies the following Cauchy-Schwarz type inequality:

$$\left|G_\lambda^{(d)}(Y_1, Y_2)\right| \leqslant \sqrt{G_\lambda^{(d)}(Y_1, Y_1^*)G_\lambda^{(d)}(Y_2^*, Y_2)}. \tag{45}$$

This inequality holds when $(Y_1, Y_2)$ belongs to domain (28). Since we assume the positivity of the spectral density in the Källén–Lehmann representation (14), eq. (45) implies that in domain (28), the absolute value of $G_\mathcal{O}(Y_1, Y_2)$ is bounded by

$$\begin{aligned}|G_\mathcal{O}(Y_1, Y_2)| &\leqslant \int_{\mathbb{R}\cup i\left(-\frac{d}{2}, \frac{d}{2}\right)} d\lambda\, \rho_\mathcal{O}(\lambda)\sqrt{G_\lambda^{(d)}(Y_1, Y_1^*)G_\lambda^{(d)}(Y_2^*, Y_2)}\\ &\leqslant \sqrt{G_\mathcal{O}(Y_1, Y_1^*)G_\mathcal{O}(Y_2^*, Y_2)},\end{aligned} \tag{46}$$

where the second inequality is a Cauchy-Schwarz inequality for the two vectors $\sqrt{G_\lambda^{(d)}(Y_1, Y_1^*)}$ and $\sqrt{G_\lambda^{(d)}(Y_2^*, Y_2)}$ with respect to the inner product $(f, g)_\varrho \equiv \int d\lambda\, \varrho_\mathcal{O}(\lambda)f(\lambda)g(\lambda)$.

The $\sigma$ variable of the two-point configurations $(Y_1, Y_1^*)$ and $(Y_2^*, Y_2)$ are computed in appendix B. The claim is that for $Y_1$ and $Y_2$ in domain (31), the range of the corresponding $\sigma$ variables is given by

$$-1 \leqslant Y_1 \cdot Y_1^*, \qquad Y_2 \cdot Y_2^* < 1, \tag{47}$$

which is exactly the range for the two-point configurations on the Euclidean sphere. Therefore, the r.h.s. of (46) is finite, meaning that the Källén-Lehmann representation (14) converges absolutely. Furthermore, the convergence is uniform as long as

$$-1 \leqslant Y_1 \cdot Y_1^*, \qquad Y_2 \cdot Y_2^* < 1-\varepsilon,$$

for any fixed positive $\varepsilon$. Together with the analyticity of each single free propagator in (14), we conclude that the two-point function is analytic in domain (28). This finishes the first step.

Now let us show that in domain (28), the range of $\sigma$ is exactly given by $\mathbb{C}\backslash[1,+\infty)$. This point was explained in [7] (see Proposition 2.2-(1) there). An easy way to see this is to consider the following class of two-point configurations

$$\begin{aligned} Y_1 &= (\sinh(t_1-i\theta_1), \cosh(t_1-i\theta_1), 0, \ldots, 0), \\ Y_2 &= (i\sin(\theta_2), \cos(\theta_2)\cos(\varphi), \cos(\theta_2)\sin(\varphi), 0, \ldots), \\ 0 &< \theta_1, \quad \theta_2 \leqslant \frac{\pi}{2}, \quad t_1, \varphi \in \mathbb{R}. \end{aligned} \tag{48}$$

This class of $(Y_1, Y_2)$ belongs to the case of (31), so it is included in the forward tube domain (28). By explicit computation, we have

$$Y_1 \cdot Y_2 \equiv U \cosh(t_1) + iV \sinh(t_1), \tag{49}$$

where $U$ and $V$ are

$$\begin{aligned} U &\equiv -\sin(\theta_1)\sin(\theta_2) + \cos(\theta_1)\cos(\theta_2)\cos(\varphi), \\ V &\equiv \cos(\theta_1)\sin(\theta_2) + \sin(\theta_1)\cos(\theta_2)\cos(\varphi). \end{aligned} \tag{50}$$

By the assumed range of $\theta_1$, $\theta_2$ and $\varphi$, the corresponding range of $(U, V)$ is given by the closed unit disc minus a point:

$$\left\{ U^2 + V^2 \leqslant 1 \right\} \backslash \{(1,0)\}. \tag{51}$$

Then by taking all possible real $t_1$ for (49), we get the range of $Y_1 \cdot Y_2$:

$$\mathbb{C}\backslash[1,+\infty), \tag{52}$$

where the interval $[1,+\infty)$ is the orbit of $U\cosh t_1 + iV\sinh t_1$ with $(U,V)=(1,0)$. This finishes the second step and the proof of corollary 3.3.

## 4 Proof of proposition 3.1

In this section, we present a comprehensive proof of proposition 3.1. Since free scalar propagators are, up to normalization, hypergeometric functions, proposition 3.1 is equivalent to the statement that

$$_2F_1\left(\frac{d}{2}+i\lambda, \frac{d}{2}-i\lambda, \frac{d+1}{2}, \frac{4\rho}{(1+\rho)^2}\right) = \sum_{n=0}^{\infty} b_n(d,\lambda)\rho^n, \qquad \lambda \in \mathbb{R} \cup i\left(-\frac{d}{2}, \frac{d}{2}\right), \tag{53}$$

with $b_n(d,\lambda) \geqslant 0$ for all $n$. The proof is structured as follows. First, we derive an explicit formula, eq. (64), for the l.h.s. of eq. (53) (section 4.1). Some intricate technical details are relegated to appendix C. Then using the established formula (64), we provide a proof of proposition 3.1 for the case of principal series in $d \geqslant 1$ (section 4.2) and the case of complementary series in $d = 1$ (section 4.3). Finally, we prove the case of complementary series in $d \geqslant 2$ through the method of dimensional reduction (section 4.4).

Figure 6: Two conformally equivalent configurations. $z$ and $\rho$ are related via (60).

(a) Configuration of (58).

(b) Configuration of (61).

## 4.1 $\rho$ expansion from CFT$_1$

In this subsection, we would like to derive the explicit form of $b_n(d, \lambda)$ in eq. (53). To begin with, we would like to introduce the following identity for hypergeometric $_2F_1$:

$$
_2F_1\left(h - \delta_1, h - \delta_2; 2h; \frac{4\rho}{(1+\rho)^2}\right)
$$
$$
= (1+\rho)^{2h}\left(\frac{1-\rho}{1+\rho}\right)^{\delta_1+\delta_2} \sum_{n=0}^{\infty} \frac{n!}{(2h)_n} a_n(h, \delta_1) a_n(h, \delta_2) \rho^n \qquad (|\rho| < 1), \quad (54)
$$

where the term $a_n$ is defined as

$$
a_n(h, \delta) \equiv \sum_{k=0}^{n} \frac{(-1)^k (h-\delta)_k (h+\delta)_{n-k}}{k!(n-k)!}. \quad (55)
$$

Our derivation of (54) is based on techniques from one-dimensional conformal field theory (CFT$_1$). We leave the technical details to appendix C. Here, we would like to briefly explain the main idea of the derivation.

The key observation is that in CFT$_1$, the conformal block of the four-point function takes the following form [58]:

$$
g_{1234,h}(z) = |z|^h \, _2F_1(h - h_{12}, h + h_{34}; 2h; z), \quad (56)
$$

where $h_i$ ($i = 1, 2, 3, 4$) denotes the scaling dimension of the external primary operators, and $h$ denotes the scaling dimension of the exchanged (internal) primary operator. Here we use $h_{ij} \equiv h_i - h_j$ for convenience. $z$ is the cross-ratio of the four-point configuration $(\tau_1, \tau_2, \tau_3, \tau_4)$, defined by

$$
z \equiv \frac{\tau_{12}\tau_{34}}{\tau_{13}\tau_{24}} \qquad (\tau_{ij} \equiv \tau_i - \tau_j). \quad (57)
$$

The expression (56) can be computed using the operator product expansion (OPE) in the following four-point configuration (see figure 6a):

$$
\tau_1 = 0, \qquad \tau_2 = z, \qquad \tau_3 = 1, \qquad \tau_4 = \infty. \quad (58)
$$

The conformal block $g_{1234,h}(z)$ is conformally invariant. Thus, in principle, we can choose another four-point configuration $(\tau_1', \tau_2', \tau_3', \tau_4')$ to compute it, if $(\tau_1', \tau_2', \tau_3', \tau_4')$ can be obtained by acting with a global conformal transformation on (58). Here we choose the following conformal transformation:

$$
\tau_i' = \frac{(1+\rho)\tau_i - 2\rho}{(1+\rho)\tau_i - 2}, \qquad i = 1, 2, 3, 4, \quad (59)
$$

with

$$
z = \frac{4\rho}{(1+\rho)^2}. \quad (60)
$$

This transformation maps the configuration (58) to the following one (see figure 6b)

$$\tau'_1 = \rho\,, \qquad \tau'_2 = -\rho\,, \qquad \tau'_3 = -1\,, \qquad \tau'_4 = 1\,. \tag{61}$$

Then, using the OPE in the configuration (61), one can show that the conformal block has the following expression:

$$g_{1234,h}\left(\frac{4\rho}{(1+\rho)^2}\right) = |4\rho|^h \left|\frac{1-\rho}{1+\rho}\right|^{h_{12}-h_{34}} \sum_{n=0}^{\infty} \frac{n!}{(2h)_n} a_n(h,h_{12})a_n(h,-h_{34})\rho^n\,, \tag{62}$$

where $a_n$ is the same as in (55).

Comparing eqs. (56) and (62) (and choosing $h_{12} = \delta_1$ and $h_{34} = \delta_2$), we obtain the identity (54).

Now we can forget about CFT and focus on (54). To apply the identity (54) to the l.h.s. of (10), we choose

$$h = \frac{d+1}{4}\,, \qquad \delta_1 = -\frac{d-1}{4} + i\lambda\,, \qquad \delta_2 = -\frac{d-1}{4} - i\lambda\,. \tag{63}$$

With this choice, the identity (54) leads to

$$\begin{aligned}
&{}_2F_1\left(\frac{d}{2} + i\lambda, \frac{d}{2} - i\lambda; \frac{d+1}{2}; \frac{4\rho}{(1+\rho)^2}\right) \\
&= \frac{(1+\rho)^d}{(1-\rho)^{\frac{d-1}{2}}} \sum_{n=0}^{\infty} \frac{n!}{\left(\frac{d+1}{2}\right)_n} a_n\left(\frac{d+1}{4}, -\frac{d-1}{4} + i\lambda\right) a_n\left(\frac{d+1}{4}, -\frac{d-1}{4} - i\lambda\right)\rho^n\,,
\end{aligned} \tag{64}$$

where the coefficients $a_n$ are defined in eq. (55).

## 4.2 Principal series in $d \geqslant 1$

The principal series corresponds to $\lambda \in \mathbb{R}$ in eq. (53). In this case, by (55), we have

$$a_n\left(\frac{d+1}{4}, -\frac{d-1}{4} + i\lambda\right) = a_n\left(\frac{d+1}{4}, -\frac{d-1}{4} - i\lambda\right)^*\,. \tag{65}$$

Consequently, the sum in the r.h.s. of (64) is a power series of $\rho$ with positive coefficients. Furthermore, the prefactor $\frac{(1+\rho)^d}{(1-\rho)^{\frac{d-1}{2}}}$ can also be expressed as a power series of $\rho$ with positive coefficients. Therefore, the whole r.h.s. of (64) is a power series of $\rho$ with positive coefficients.[5]

This completes the proof of proposition 3.1 for the case of the principal series in $d \geqslant 1$.

*Remark* 4.1. Eq. (65) does not hold for $\lambda \in i\left(-\frac{d}{2}, \frac{d}{2}\right)$. The argument presented here thus fails in the case of the complementary series. For example, in $d = 1$, the identity (64) reduces to

$$_2F_1\left(\frac{1}{2} + i\lambda, \frac{1}{2} - i\lambda; 1; \frac{4\rho}{(1+\rho)^2}\right) = (1+\rho)\sum_{n=0}^{\infty}(-1)^n a_n\left(\frac{1}{2}, i\lambda\right)^2 \rho^n\,. \tag{66}$$

Here we used the identity $a_n(h, \delta) = (-1)^n a_n(h, -\delta)$ by definition (55). Without the $(1 + \rho)$ prefactor, the r.h.s. is not a positive sum.

Therefore, the case of the complementary series needs to be treated on its own, with a different approach.

---

[5] To be more precise, the coefficient $b_n(d, \lambda)$ in eq. (53) is given by (13).

### 4.3 Complementary series in $d = 1$

Here we will present a proof of proposition 3.1 for the complementary series in $d = 1$, i.e. the range $\lambda \in i\left(-\frac{1}{2}, \frac{1}{2}\right)$. For convenience, let us introduce the notation

$$\Delta = \frac{1}{2} + i\lambda, \qquad \bar{\Delta} = 1 - \Delta. \tag{67}$$

It follows that $\lambda \in i\left(-\frac{1}{2}, \frac{1}{2}\right)$ corresponds to $\Delta \in (0, 1)$.

Plugging $d = 1$ in the identity (64) yields

$$_2F_1\left(\Delta, 1 - \Delta; 1; \frac{4\rho}{(1+\rho)^2}\right) = \sum_{n=0}^{\infty}\left[\alpha_n(\Delta)\alpha_n(\bar{\Delta}) + \alpha_{n-1}(\Delta)\alpha_{n-1}(\bar{\Delta})\right]\rho^n, \tag{68}$$

where $\alpha_n(\Delta) \equiv a_n(\frac{1}{2}, \Delta - \frac{1}{2})$ (c.f. eq. (55)), i.e.

$$\alpha_n(\Delta) = \sum_{k=0}^{n} \frac{(-1)^k(\bar{\Delta})_k(\Delta)_{n-k}}{k!(n-k)!}. \tag{69}$$

By definition, we have $\alpha_n(\bar{\Delta}) = (-1)^n\alpha_n(\Delta)$ and hence the coefficient of $\rho^n$ in (68) equals the product $c_n(\Delta)c_n(\bar{\Delta})$, where $c_n(\Delta) \equiv \alpha_n(\Delta) + \alpha_{n-1}(\Delta)$. To proceed further, we need a simple lemma.

**Lemma 4.2.** $n!\alpha_n(\Delta)$ *is equal to the n-th derivative of* $(1+x)^{-\bar{\Delta}}(1-x)^{-\Delta}$*, evaluated at* $x = 0$*.*

*Proof.* The lemma follows from a direct computation of derivatives:

$$\partial_x^n\left[(1+x)^{-\bar{\Delta}}(1-x)^{-\Delta}\right] = \sum_{k=0}^{n}\binom{n}{k}\partial_x^k(1+x)^{-\bar{\Delta}}\partial_x^{n-k}(1-x)^{-\Delta}$$

$$= \sum_{k=0}^{n}\binom{n}{k}\frac{(-1)^k(\bar{\Delta})_k(\Delta)_{n-k}}{(1+x)^{\bar{\Delta}+k}(1-x)^{\Delta+n-k}}. \tag{70}$$

Taking $x = 0$, we obtain $\partial_x^n\left[(1+x)^{-\bar{\Delta}}(1-x)^{-\Delta}\right]\Big|_{x=0} = n!\alpha_n(\Delta)$. $\qquad\square$

Defining $\phi_\Delta(x) = \left(\frac{1+x}{1-x}\right)^\Delta$ and using this lemma, we find

$$\alpha_n(\Delta) = \frac{1}{n!}\partial_x^n\big|_{x=0}\left(\frac{1}{1+x}\phi_\Delta(x)\right) = \sum_{l=0}^{n}\frac{(-1)^{n-l}}{l!}\partial^l\phi_\Delta(0), \tag{71}$$

which yields $c_n(\Delta) = \frac{1}{n!}\partial^n\phi_\Delta(0)$. Altogether, the $\rho$ expansion (68) becomes

$$_2F_1\left(\Delta, 1 - \Delta; 1; \frac{4\rho}{(1+\rho)^2}\right) = \sum_{n=0}^{\infty}\frac{\partial^n\phi_\Delta(0)}{n!}\frac{\partial^n\phi_{\bar{\Delta}}(0)}{n!}\rho^n. \tag{72}$$

The last step is to show that $\partial^n\phi_\Delta(0) > 0$ for all $n \in \mathbb{N}$ and $\Delta \in (0, 1)$. When $n = 0$, this holds trivially since $\phi_\Delta(0) = 1$. When $n \geqslant 1$, it is a result of the following integral representation:

**Lemma 4.3.** *For any positive integer n and* $\Delta \in (0, 1)$*, we have*

$$\frac{\partial^n\phi_\Delta(0)}{n!} = \frac{\sin(\pi\Delta)}{\pi}\int_1^{\infty}\frac{dr}{r^{n+1}}\left[\left(\frac{r+1}{r-1}\right)^\Delta - (-)^n\left(\frac{r-1}{r+1}\right)^\Delta\right]. \tag{73}$$

*Because the integrand is manifestly positive for* $\Delta \in (0, 1)$*, the n-th derivative* $\partial^n\phi_\Delta(0)$ *is also positive.*

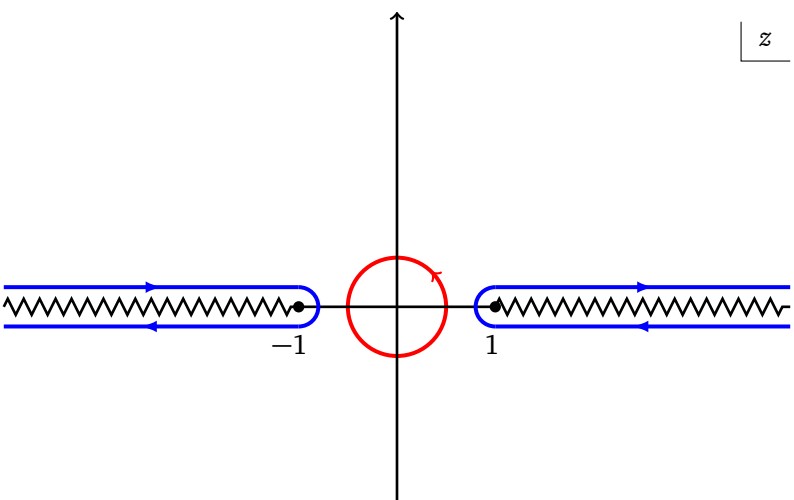

Figure 7: Branch cuts of $\phi_\Delta(z)$ in the $z$-plane and contour deformations of the integral (74).

*Proof.* Consider $\phi_\Delta(z)$ as a complex function of $z \in \mathbb{C}$. Since $\phi_\Delta(z)$ is holomorphic in the domain $|z| < 1$, its derivative at $z = 0$ admits an integral representation

$$\frac{\partial^n \phi_\Delta(0)}{n!} = \oint_{C_0} \frac{dz}{2\pi i} \frac{\phi_\Delta(z)}{z^{n+1}}, \tag{74}$$

where the contour $C_0$ is contained in the unit disk, as shown by the red circle in figure 7. On the other hand, since $\phi_\Delta(z)$ is holomorphic in the cut plane $\mathbb{C}\backslash(-\infty, -1] \cup [1, \infty)$ and bounded by 1 at $\infty$, we can deform the contour $C_0$ such that it runs along the branch cuts of $\phi_\Delta(z)$. The deformed contour is shown in blue in figure 7. The new contour integral relates $\partial^n \phi_\Delta(0)$ and the discontinuity of $\phi_\Delta(z)$ at the branch cuts[6]

$$\frac{\partial^n \phi_\Delta(0)}{n!} = \frac{1}{2\pi i} \int_1^\infty \frac{dr}{r^{n+1}} \left( \text{Disc}[\phi_\Delta](r) - (-)^n \text{Disc}[\phi_\Delta](-r) \right), \tag{75}$$

where $\text{Disc}[f](r) \equiv \lim_{\epsilon \to 0^+} f(r + i\epsilon) - f(r - i\epsilon)$. By carefully analyzing the discontinuity, we find

$$\text{Disc}[\phi_\Delta](\pm r) = 2i \sin(\Delta \pi) \left( \frac{r+1}{r-1} \right)^{\pm \Delta}, \tag{76}$$

when $r > 1$, and thus we recover eq. (73). $\square$

Combining eq. (72) and lemma 4.3 we have shown that complementary series propagators in $\text{dS}_2$ have the $\rho$ expansion (53) with

$$b_n(1, \lambda) = \frac{\partial^n \phi_\Delta(0)}{n!} \frac{\partial^n \phi_{\bar\Delta}(0)}{n!} > 0. \tag{77}$$

This proves the complementary series case of proposition 3.1 when $d = 1$, i.e. $b_n(1, \lambda) > 0$ for any $n \in \mathbb{N}$ and $\lambda \in i\left(-\frac{1}{2}, \frac{1}{2}\right)$.

---

[6]Because of $0 < \Delta < 1$, we can neglect the small half-circles around the two branch points $z = \pm 1$.

### 4.4  Complementary series in $d \geqslant 2$

In this section, we will prove proposition 3.1 for the case of $\lambda \in i\left(-\frac{d}{2}, \frac{d}{2}\right)$, the complementary series, in any dimension $d$ by using dimensional reduction. It is worth noting that in section 4.3, we demonstrated that a complementary series free propagator (which is proportional to $_2F_1$) in $d = 1$ possesses a positive series expansion in the variable $\rho$. The main idea in this section is then to prove that in $d \geqslant 2$, free scalar propagators in the principal or complementary series have a positive Källén–Lehmann decomposition into free scalar propagators in the principal or complementary series in $d = 1$, thus inheriting the property of having a positive series expansion in the $\rho$ variable. The underlying concept behind this process of dimensional reduction is that any unitary QFT in $\mathrm{dS}_{d+1}$ can be regarded as a unitary QFT in $\mathrm{dS}_2$ when we confine the domain of the correlation functions to a $\mathrm{dS}_2$ slice within $\mathrm{dS}_{d+1}$.

The proof is going to be split in two parts. First, we consider free propagators $G_\lambda^{(d)}(\sigma)$ with $|\mathrm{Im}(\lambda)| < \frac{d-1}{2}$, which includes all the principal series and part of the complementary series, and then the case $\frac{d-1}{2} < |\mathrm{Im}(\lambda)| < \frac{d}{2}$, covering the rest of the complementary series.[7]

### 4.4.1  Case $|\mathrm{Im}(\lambda)| < \frac{d-1}{2}$

Our starting point is the fact that the free scalar propagators of principal series in $\mathrm{dS}_2$,

$$G_\nu^{(1)}(\sigma) = \frac{\Gamma(\frac{1}{2} \pm i\nu)}{4\pi} \mathbf{F}\left(\frac{1}{2} + i\nu, \frac{1}{2} - i\nu; 1; \frac{1+\sigma}{2}\right), \qquad \nu \in \mathbb{R}, \tag{78}$$

form an orthogonal basis for square-integrable functions over the interval $\sigma \in (-\infty, -1]$,[8] where the orthogonality relation is given by

$$\int_{-\infty}^{-1} d\sigma \, G_\lambda^{(1)}(\sigma) G_\nu^{(1)}(\sigma) = \frac{\delta(\lambda - \nu) + \delta(\lambda + \nu)}{8\nu \sinh(2\pi\nu)}. \tag{79}$$

The $\mathrm{dS}_{d+1}$ propagators $G_\lambda^{(d)}(\sigma)$ are regular at $\sigma = -1$. Furthermore, in the vicinity of $\sigma = -\infty$, they have the following asymptotic behavior:

$$G_\lambda^{(d)}(\sigma) = \frac{\Gamma(-2i\lambda)\Gamma\left(\frac{d}{2} + i\lambda\right)}{(4\pi)^{\frac{d+1}{2}}\Gamma\left(\frac{1}{2} - i\lambda\right)}(-\sigma)^{-\frac{d}{2} - i\lambda}\left[1 + O\left(|\sigma|^{-1}\right)\right] + (\lambda \to -\lambda). \tag{80}$$

Consequently, the condition for square integrability over $\sigma \in (-\infty, -1]$ is met when:

$$|\mathrm{Im}(\lambda)| < \frac{d-1}{2}. \tag{81}$$

Under the condition (81), we can express $G_\lambda^{(d)}(\sigma)$ as:

$$G_\lambda^{(d)}(\sigma) = \int_{\mathbb{R}} d\nu \, \varrho_\lambda^{\mathcal{P}}(\nu) G_\nu^{(1)}(\sigma), \qquad \sigma \in (-\infty, -1]. \tag{82}$$

To determine the spectral density $\varrho_\lambda^{\mathcal{P}}(\nu)$, we use (79) and obtain

$$\varrho_\lambda^{\mathcal{P}}(\nu) = \frac{\Gamma(\frac{d}{2} \pm i\lambda)}{(4\pi)^{\frac{d-1}{2}}\Gamma(\pm i\nu)} \int_0^\infty dz \, \mathbf{F}\left(\frac{1}{2} + i\nu, \frac{1}{2} - i\nu; 1; -z\right) \mathbf{F}\left(\frac{d}{2} + i\lambda, \frac{d}{2} - i\lambda; \frac{d+1}{2}; -z\right), \tag{83}$$

where we changed variables to $z = -\frac{1+\sigma}{2}$. After some technical steps which are detailed in appendix D.1, we obtain

---

[7]By continuity, the same conclusion will hold for the critical case $|\mathrm{Im}(\lambda)| = \frac{d-1}{2}$.

[8]The validation for this claim is provided in section II of [59] where the analysis was conducted on the Euclidean hyperbolic surface, also known as Euclidean AdS (EAdS). In the context of the two-point configuration in EAdS, the range for $\sigma$ is $\sigma \in (-\infty, -1]$.

$$\varrho_\lambda^{\mathcal{P}}(\nu) = \frac{\nu \sinh \pi \nu}{8\pi^{\frac{d+3}{2}}\Gamma\left(\frac{d-1}{2}\right)}\prod_{\pm,\pm}\Gamma\left(\frac{d-1}{4}\pm i\frac{\nu}{2}\pm i\frac{\lambda}{2}\right). \tag{84}$$

Notice that $\varrho_\lambda^{\mathcal{P}}(\nu)$ is positive in the ranges of interest $\lambda \in \mathbb{R}\cup i(-\frac{d-1}{2},\frac{d-1}{2})$ and $\nu \in \mathbb{R}$.

We thus see that a principal series or complementary series propagator satisying (81) in $\mathrm{dS}_{d+1}$ only contains states in the two-dimensional principal series, when restricted to a $\mathrm{dS}_2$ slice. Moreover, given that we proved that $G_\lambda^{(1)}(\sigma)$ has a positive series expansion in the $\rho$ variable, and given the positivity of $\varrho_\lambda^{\mathcal{P}}(\nu)$, we can state that $G_\lambda^{(d)}(\sigma)$ with $\lambda \in \mathbb{R}\cup i(-\frac{d-1}{2},\frac{d-1}{2})$ also has a positive series expansion in $\rho$.

This finishes the proof for the case of $\lambda \in i(-\frac{d-1}{2},\frac{d-1}{2})$.

### 4.4.2   Case $\frac{d-1}{2} < |\mathrm{Im}(\lambda)| < \frac{d}{2}$

We now aim to extend the dimensional reduction formula (82) to encompass the regime $\frac{d-1}{2} < |\mathrm{Im}(\lambda)| < \frac{d}{2}$. This range includes the remaining part of the complementary series. It is worth noting that the free propagator $G_\lambda^{(d)}(\sigma)$, as defined in (5), is analytic in $\lambda$ in the domain

$$\mathrm{Re}(\lambda)\in\mathbb{R}, \qquad |\mathrm{Im}(\lambda)| < \frac{d}{2}. \tag{85}$$

Our goal in this subsection is to reformulate the integral (82) in such a way that it incorporates this essential analyticity property of $G_\lambda^{(d)}(\sigma)$ with respect to $\lambda$. Here we are going to take a heuristic approach, while we leave the rigorous approach and most of the details to appendix D.2.

Let us start from (82). The spectral density (83) has poles at

$$\nu = \pm\lambda \pm i\left(\frac{d-1}{2}+2n\right) \qquad (n=0,1,2,\ldots). \tag{86}$$

When continuing $\lambda$ above $|\mathrm{Im}(\lambda)| = \frac{d-1}{2}$, two of these poles, corresponding to $n=0$, will cross the integration contour over real axis of $\nu$. To maintain analyticity, the residues on their positions need to be added in order to obtain the full answer. For imaginary $\lambda$, these two poles correspond to one specific representation in the $\mathrm{dS}_2$ complementary series. The result, for the range $\mathrm{Im}(\lambda)\in\left(-\frac{d}{2},\frac{d}{2}\right)\backslash\{\pm\frac{d-1}{2}\}$,[9] can be written as

$$G_\lambda^{(d)}(\sigma) = \int_\mathbb{R} d\nu\, \varrho_\lambda^{\mathcal{P}}(\nu)G_\nu^{(1)}(\sigma) + \left[\Theta\left(\mathrm{Im}(\lambda)-\frac{d-1}{2}\right)\varrho_\lambda^{\mathcal{C}}\,G_{\lambda-i\frac{d-1}{2}}^{(1)}(\sigma) + (\lambda\leftrightarrow-\lambda)\right], \tag{87}$$

where $\Theta$ is a step function and

$$\varrho_\lambda^{\mathcal{C}} = \frac{\pi^{\frac{1-d}{2}}\Gamma(-i\lambda)}{\Gamma\left(-i\lambda-\frac{d-1}{2}\right)}. \tag{88}$$

The density $\varrho_\lambda^{\mathcal{P}}(\nu)$ is given by eq. (84). We can thus state that a $\mathrm{dS}_{d+1}$ free propagator in the principal or complementary series only includes principal series and at most one UIR in the complementary series, when reduced to $\mathrm{dS}_2$. Importantly, the spectral densities are positive in this reduction, so that indeed the property of having a positive series expansion in the $\rho$ variable is inherited by the higher dimensional propagators. This concludes the proof of proposition 3.1.

---

[9]The function $G_\lambda^{(d)}$ is continuous at $\mathrm{Im}(\lambda)=\pm\frac{d-1}{2}$, and its value at that point is equal to the limit from below and from above of equation (87).

Before moving to the discussion section, let us make some remarks about this decomposition:

- In the case of $d = 1$, eq. (84) simplifies to

$$\varrho_\lambda^{\mathcal{P}}(\nu) = \begin{cases} \frac{1}{2}\delta(\nu+\lambda) + \frac{1}{2}\delta(\nu-\lambda), & \lambda \in \mathbb{R}, \\ 0, & \lambda \notin \mathbb{R}. \end{cases} \tag{89}$$

  Subsequently, eqs. (82) and (87) become the trivial equation

$$G_\lambda^{(1)}(\sigma) = G_\lambda^{(1)}(\sigma).$$

- The absence of $SO(2,1)$ discrete series in the dimensional reduction (87) has a purely group theoretical explanation. More precisely, given a scalar principal or complementary series representation $\mathcal{F}_\Delta$ of $SO(d+1,1)$, it can be shown that the restriction of $\mathcal{F}_\Delta$ to the $SO(d,1)$ subgroup consists of principal and complementary series of $SO(d,1)$. The detailed proof of this proposition is given in appendix E.

- In appendix D.2, we derive (87) in a more rigorous way, being careful about the analytic continuation and detailing every step.

- In appendix D.3, we show that sending the radius of de Sitter to infinity, these decompositions reduce to their correct analogues in flat space. In particular, in the flat space limit, the poles of $\rho_\lambda^{\mathcal{P}}(\nu)$ condense and form a branch cut.

# 5 Discussion

The main outcome of this work is proposition 3.1, stating that free propagators in de Sitter have a series expansion in the $\rho$ variable which has only positive coefficients, and corollaries 3.2 and 3.3, which state non-perturbatively that assuming a Källén-Lehmann decomposition of the form (14), the Wightman two-point function $G_{\mathcal{O}}(\sigma)$ of any scalar operator $\mathcal{O}$ has a positive series expansion in $\rho$ and is analytic in the "maximal analyticity" domain, corresponding to $\sigma \in \mathbb{C}\backslash[1,\infty)$. Finally, we elaborated on the fact that analytic continuation between the sphere, de Sitter and EAdS happen through paths that are included in this domain of analyticity. Here we mention some remaining open questions that would be interesting to explore in the future

- What is the physical meaning of the radial variable $\rho$? Usually, when an observable can be expanded as a sum of positive terms, there is a conceptual meaning to the expansion, and a physical principle dictating the positivity. For example, for CFT four-point functions, the coefficients of the $\rho$ expansion have the physical meaning of the inner products of the states in the Hilbert space, so the positivity of the coefficients naturally follows from positivity of the inner product [42].

  At this moment, for de Sitter QFT we lack the intuition to explain why, in simple terms and beyond the mathematical proof we exposed in section 4, the coefficients in (10) need to be positive.

- What is the analytic structure of a two-point function beyond the first sheet, in general? The hypergeometric functions which appear as blocks in the Källén-Lehmann decomposition (14) have infinite sheets, which are accessed by crossing the cut. It would be interesting to understand whether their analytic structure is inherited by two-point functions non-perturbatively even beyond the first sheet.

- The generalization of the results presented in this paper to the case of two-point functions of operators with spin is not completely trivial. In fact, in the index-free formalism of [17, 20], free propagators of spinning fields are combinations of scalar propagators multiplied by polynomials of $\{\sigma, (Y_1 \cdot W_2)(Y_2 \cdot W_1), (W_1 \cdot W_2)\}$, where $W_1$ and $W_2$ are auxiliary vectors encoding the spin of fields at $Y_1$ and $Y_2$ respectively.[10] Since $\sigma = \frac{8\rho}{(1+\rho)^2} - 1$, it is not immediate to prove analytically that spinning two-point functions also have a positive series expansion in the radial variable. Nevertheless, let us mention that numerical checks suggest that the coefficients of the tensor structures $\{(Y_1 \cdot W_2)(Y_2 \cdot W_1), (W_1 \cdot W_2)\}$ for free fields of spin 1 and 2 do indeed have a positive $\rho$ expansion.

- Is it possible to leverage the positivity of the series coefficients proved in corollary 3.2 in a numerical setup to constrain observables in QFT in dS? In [60] we define c-functions and sum rules to extract the central charges at the endpoints of RG flows in $dS_2$. In particular, a certain sum rule relates $c^{\text{UV}}$ to an integral over the bulk two-point function of the trace of the stress tensor $\Theta$. Using corollary 3.2, we can thus write the UV central charge as a sum over positive coefficients. Is it possible to find an independent physical constraint on the coefficients of the $\rho$ expansion of $\Theta$ and find a universal minimum to $c^{\text{UV}}$?

- What is the analytic structure of an $n$-point function? In flat space, time translation symmetry, reflection positivity and polynomial boundedness are enough to prove analyticity for higher-point functions. In de Sitter and on the sphere, instead, there is no time translation symmetry. The approach we take in this paper is based on the structure of the Källén-Lehmann representation, and thus only apply to two-point functions. Currently we do not know how to prove the analyticity of higher-point functions.

# Acknowledgments

We are grateful to Tarek Anous, Frederik Denef, Victor Gorbenko, Shota Komatsu, Mehrdad Mirbabayi, Joao Penedones, Guilherme Pimentel, Fedor Popov, Kamran Salehi Vaziri, Antoine Vuignier and Alexander Zhiboedov for useful discussions. ML and JQ thank CERN for hospitality during the conference "Cosmology, Quantum Gravity, and Holography: the Interplay of Fundamental Concepts".

**Funding information** ML is supported by the Simons Foundation grant 488649 (Simons Collaboration on the Nonperturbative Bootstrap) and the Swiss National Science Foundation through the project 200020_197160 and through the National Centre of Competence in Research SwissMAP. JQ is supported by Simons Collaboration on Confinement and QCD Strings and by the National Centre of Competence in Research SwissMAP. ZS is supported by the US National Science Foundation under Grant No. PHY2209997 and the Gravity Initiative at Princeton University.

---

[10]More precisely, $W_k$ $(k = 1, 2)$ is a tangential and null vector in the embedding space, satisfying $Y_k \cdot W_k = W_k \cdot W_k = 0$.

# A  No exceptional or discrete series states in scalar two-point functions

Let us elaborate on the absence of contributions from the exceptional series type I in (14) (the arguments would be analogous for the discrete series in $dS_2$. See appendix E.1 for a quick review of all scalar UIRs). When going through the derivation of the Källén-Lehmann decomposition in [17], it was argued that the solutions to the Casimir equation for objects such as

$$\langle \Omega | \mathcal{O}(Y_1) \mathbb{1}_{\mathcal{V}_{p,0}} \mathcal{O}(Y_2) | \Omega \rangle, \tag{A.1}$$

are either growing polynomially at infinite separation, or have cuts for space-like two-point configurations (specifically at $\sigma \in (-\infty, -1]$), where $\mathbb{1}_{\mathcal{V}_{p,0}}$ denotes a projector to the UIR $\mathcal{V}_{p,0}$. From the point of view of QFT in de Sitter we are forced to exclude the contributions which grow polynomially, while we cannot completely exclude the possibility that contributions which diverge at $\sigma = -1$ associated to different $p$ conspire to cancel the overall singularity, since the sign of the divergence depends on $p$. In other words, we cannot rigorously exclude the possibility that some very complicated operator $\mathcal{O}$ creates states in the exceptional series that sum up to a physically admissible two-point function.

Nevertheless, there is no example in the literature of a scalar two-point function which includes discrete or exceptional series states in its Källén-Lehmann representation. In [17], scalar two-point functions were studied in CFT, weakly coupled $\phi^4$ theory and composite operators in free theory. All of these examples only include principal and complementary series contributions. The most striking case is probably that of the two-point function of $\phi^2$, where $\phi$ is a free massive scalar. In [61–63], it was shown that the decomposition of the tensor product of two states in the principal series in $dS_2$ includes states in the discrete series. Nevertheless, the Källén-Lehmann decomposition of $\langle \phi^2(Y_1)\phi^2(Y_2) \rangle$ does not show the appearance of any such state. Inspired by the plethora of examples and by the required conspiracy to cancel unphysical singularities in de Sitter, we thus phrase the following conjecture:

**Conjecture:** *In a unitary QFT in de Sitter, no scalar local operator $\mathcal{O}(Y)$, acting on the Bunch-Davies vacuum, can create states in the exceptional series $\mathcal{V}_{p,0}$ in $dS_{d+1}$ and in the discrete series $\mathcal{D}_p^{\pm}$ in $dS_2$.*

For de Sitter QFT, correlation functions in the Bunch-Davies vacuum are by definition analytic continuations of their analogues on the sphere. Thus the above conjecture can be rephrased as follows

**Conjecture:** *Consider a scalar two-point function on $S^{d+1}$ that is regular when the two points are not coincident, $SO(d+2)$ invariant and is reflection positive. Then, it has a Källén-Lehmann representation of the form (14) and contains no representations in the exceptional series $\mathcal{V}_{p,0}$ in $S^{d+1}$ and in the discrete series $\mathcal{D}_p^{\pm}$ in $S^2$.*

Let us make some remarks about these conjectures

- In contrast with scalars, operators with spin *can* create such states. An example is the CFT conserved current that was previously explored in section 5.2.2 in [17], which in $dS_2$ creates states in the discrete series $\mathcal{D}_1^{\pm}$. The general statement is that a spin $J$ operator can create discrete series states with $p = 1, \dots, J$. The blocks in that case will decay at large distances and be free of branch points at space-like separation.

- In [64], the authors show that it seems to be possible to construct scalar two-point functions that decompose into states in $\mathcal{D}_p^{\pm}$ in $dS_2$. In their construction, they subtract

an SO(3)-invariant singular term, which renders the modified two-point function well-behaved at the antipodal singularity ($\sigma = -1$). Nevertheless, it is important to note that the modified two-point functions do not satisfy the condition of our conjecture because they lack reflection positivity on the sphere.

• The presence of type II exceptional series (denoted as $\mathcal{U}_{s,t}$ in [49]) and higher dimensional discrete series in the Källén-Lehmann decomposition of scalar two-point functions in $dS_{d+1}$ is directly forbidden by symmetry (see [17] for more discussions on this), and is thus *not* a conjecture.

## B  Computing $\sigma$-variable for symmetric two-point configurations

In this appendix, we would like to compute the $\sigma$-variable for the following symmetric two-point configurations:

$$Y_1 = Y_2^*\,, \tag{B.1}$$

where $Y_1$ and $Y_2$ is in the complex embedding space $\mathbb{C}^{1,d+1}$ and satisfying

$$Y_1^2 = 1\,, \qquad \mathrm{Im}(Y_1) \in V_+\,. \tag{B.2}$$

Here we have set the de Sitter radius $R = 1$ for convenience.

In this case, we have

$$\sigma \equiv Y_1 \cdot Y_2 = (\mathrm{Re}(Y_1))^2 + (\mathrm{Im}(Y_1))^2\,. \tag{B.3}$$

Expanding the first condition of (B.2), we get

$$(\mathrm{Re}(Y_1))^2 - (\mathrm{Im}(Y_1))^2 + i(\mathrm{Re}(Y_1) \cdot \mathrm{Im}(Y_1)) = 1\,, \tag{B.4}$$

so

$$(\mathrm{Re}(Y_1))^2 - (\mathrm{Im}(Y_1))^2 = 1\,, \qquad \mathrm{Re}(Y_1) \cdot \mathrm{Im}(Y_1) = 0\,. \tag{B.5}$$

The second condition of (B.2) says that $\mathrm{Im}(Y_1)$ is time-like, so $(\mathrm{Im}(Y_1))^2 < 0$ and $\mathrm{Re}(Y_1)$ is either space-like or equal to zero, i.e., $(\mathrm{Re}(Y_1))^2 \geqslant 0$ (as a consequence of the second equation of (B.5)). Therefore, by (B.3) we have

$$-1 \leqslant \sigma < 1\,. \tag{B.6}$$

The lower bound follows from $\sigma = 2(\mathrm{Re}(Y_1))^2 - 1 \geqslant -1$, and is saturated when $\mathrm{Re}(Y_1) = 0$. The upper bound follows from $\sigma = 1 + 2(\mathrm{Im}(Y_1))^2 < 1$, and can be approached by taking the limit $\mathrm{Im}(Y_1)^2 \to 0$. This two-sided bound tells us that for configurations satisfying conditions (B.1) and (B.2), the range of $\sigma$ is exactly the same as the range of the one on the Euclidean sphere.

Now let us do the explicit computation of $\sigma$ in two coordinate systems: global coordinates and planar coordinates.

In global coordinates, we have

$$Y_1^0 = \sinh(t_1 + i\theta_1)\,, \qquad Y_1^a = \cosh(t_1 + i\theta_1)\Omega_1^a\,. \tag{B.7}$$

For the purpose of analytic continuation from Euclidean sphere to dS, we focus on the regime with real $t_1$, $\theta_1$ and $\Omega_1$, with the extra constraint $0 < \theta_1 \leqslant \frac{\pi}{2}$. Then

$$\sigma \equiv Y_1 \cdot Y_2 = Y_1 \cdot Y_1^* = \cos(2\theta_1)\,. \tag{B.8}$$

We see that $\sigma$ only depends on $\theta_1$ in this case.

In planar coordinates we have

$$Y_1^0 = \frac{(\eta_1 - iz_1)^2 - \mathbf{y}^2 - 1}{2(\eta_1 - iz_1)} , \qquad Y_1^i = -\frac{y_1^i}{\eta_1 - iz_1} , \qquad Y^{d+1} = \frac{(\eta_1 - iz_1)^2 - \mathbf{y}_1^2 + 1}{2(\eta_1 - iz_1)} . \tag{B.9}$$

For the purpose of analytic continuation from EAdS to dS, we focus on the regime with real $\eta_1, z_1$ and $\mathbf{y}_1$, with the extra constraints $-\eta_1, z_1 > 0$. Then

$$\sigma \equiv Y_1 \cdot Y_2 \equiv Y_1 \cdot Y_1^* = \frac{\eta_1^2 - z_1^2}{\eta_1^2 + z_1^2} . \tag{B.10}$$

## C   Conformal blocks in CFT$_1$

We consider conformal field theory in the one-dimensional Euclidean space (CFT$_1$). A CFT$_1$ is defined by a collection of correlation functions of so-called *primary operators*

$$\langle \mathcal{O}_1(\tau_1) \mathcal{O}_2(\tau_2) \cdots \mathcal{O}_n(\tau_n) \rangle .$$

Given any global conformal transformation

$$f(\tau) = \frac{a\tau + b}{c\tau + d} , \qquad \begin{pmatrix} a & b \\ c & d \end{pmatrix} \in GL(2; \mathbb{R}) , \tag{C.1}$$

these correlation functions satisfy conformal invariance, meaning that

$$\langle \mathcal{O}_1(\tau_1) \mathcal{O}_2(\tau_2) \cdots \mathcal{O}_n(\tau_n) \rangle = \langle \mathcal{O}'_1(\tau_1) \mathcal{O}'_2(\tau_2) \cdots \mathcal{O}'_n(\tau_n) \rangle ,$$
$$\mathcal{O}'_i(\tau) := [\partial_\tau f(\tau)]^{-h_i} \mathcal{O}_i \left( f^{-1}(\tau) \right) . \tag{C.2}$$

A CFT$_1$ is determined by the following data:

- (Spectrum) The scaling dimensions $h_i$ of the primary operators. We choose a basis of primary operators $\mathcal{O}_i$ with the following normalization:

$$\langle \mathcal{O}_i(\tau_1) \mathcal{O}_j(\tau_2) \rangle = \frac{\delta_{ij}}{|\tau_{12}|^{2h_i}} , \tag{C.3}$$

  where $\tau_{ij} \equiv \tau_i - \tau_j$.

- (Dynamics) Three-point functions of primary operators:

$$\langle \mathcal{O}_i(\tau_1) \mathcal{O}_j(\tau_2) \mathcal{O}_k(\tau_3) \rangle = \frac{C_{ijk}}{|\tau_{12}|^{h_i+h_j-h_k} |\tau_{23}|^{h_j+h_k-h_i} |\tau_{13}|^{h_i+h_k-h_j}} . \tag{C.4}$$

In other words, a CFT$_1$ is determined by a collection of quantum numbers $\{h_i\}$ and "couplings" $\{C_{ijk}\}$. In principle, all the higher-point functions can be computed from these data using the operator product expansion (OPE):

$$\mathcal{O}_i(\tau_1) \mathcal{O}_j(\tau_2) = \sum_k A_{ijk}(\tau_1, \tau_2, \tau_0, \partial_0) \mathcal{O}_k(\tau_0) , \tag{C.5}$$

where the sum is over all primary operators, and $A_{ijk}$ is fully determined by $\{h_i\}$ and $\{C_{ijk}\}$. For the OPE to be convergent, $\tau_0$ is chosen such that $|\tau_1 - \tau_0|$ and $|\tau_2 - \tau_0|$ are smaller than other $|\tau_i - \tau_0|$'s in the correlation function.

For the purpose of this work, let us consider the four-point function of primary operators. By conformal invariance, it has the following form:

$$\langle \mathcal{O}_1(\tau_1)\mathcal{O}_2(\tau_2)\mathcal{O}_3(\tau_3)\mathcal{O}_4(\tau_4)\rangle = \frac{1}{|\tau_{12}|^{h_1+h_2}|\tau_{34}|^{h_3+h_4}}\left|\frac{\tau_{14}}{\tau_{24}}\right|^{h_{21}}\left|\frac{\tau_{14}}{\tau_{13}}\right|^{h_{34}}g_{1234}(z), \quad (C.6)$$

where $\tau_{ij} \equiv \tau_i - \tau_j$, $h_{ij} \equiv h_i - h_j$, and $z$ represents the cross-ratio defined as

$$z := \frac{\tau_{12}\tau_{34}}{\tau_{13}\tau_{24}}. \quad (C.7)$$

By OPE (C.5), the four-point function $\langle \mathcal{O}_1(\tau_1)\mathcal{O}_2(\tau_2)\mathcal{O}_3(\tau_3)\mathcal{O}_4(\tau_4)\rangle$ has an expansion in terms of conformal partial waves. Schematically, we have:

$$\langle \mathcal{O}_1(\tau_1)\mathcal{O}_2(\tau_2)\mathcal{O}_3(\tau_3)\mathcal{O}_4(\tau_4)\rangle = \sum_k \quad , \quad (C.8)$$

where the sum is over all primary operators. Each term in the sum (C.8) is conformally invariant, and takes on a similar expression to (C.6):

$$= \frac{C_{12k}C_{34k}}{|\tau_{12}|^{h_1+h_2}|\tau_{34}|^{h_3+h_4}}\left|\frac{\tau_{14}}{\tau_{24}}\right|^{h_{21}}\left|\frac{\tau_{14}}{\tau_{13}}\right|^{h_{34}}g_{1234,h_k}(z). \quad (C.9)$$

Here, $C_{12k}$ and $C_{34k}$ are the constant factors of three-point functions (C.4). By (C.6), (C.8) and (C.9), we express the conformally invariant part $g_{1234}(z)$ of the four-point function as a sum of conformal blocks:

$$g_{1234}(z) = \sum_k C_{12k}C_{34k}\,g_{1234,h_k}(z). \quad (C.10)$$

Here again, the sum is over primary operators $\mathcal{O}_k$. The conformal block, $g_{1234,h_k}(z)$, is uniquely determined by five quantum numbers: $h_i$ ($i = 1, 2, 3, 4$) and $h_k$.

Now let us derive the explicit form of $g_{1234,h}(z)$. Let $\mathcal{O}$ be a primary operator. We choose $\tau_0 = 0$ in (C.5), then the $\mathcal{O}$-channel of (C.5) can be written as

$$\mathcal{O}_1(\tau_1)\mathcal{O}_2(\tau_2) \overset{\mathcal{O}}{=} C_{12\mathcal{O}}\sum_{n=0}^{\infty}B_n(\tau_1,\tau_2)\partial^n\mathcal{O}(0). \quad (C.11)$$

Here, $\overset{\mathcal{O}}{=}$ signifies the contribution from $\mathcal{O}$ and its derivatives.

To compute the conformal block $g_{1234,h}$ using OPE, we first need to compute the OPE kernel $B_n(\tau_1,\tau_2)$. The computation of $B_n(\tau_1,\tau_2)$ can be done by analyzing the three-point function $\langle \mathcal{O}_1(\tau_1)\mathcal{O}_2(\tau_2)\mathcal{O}(L)\rangle$ in the regime where

$$|\tau_1|,\, |\tau_2| < L. \quad (C.12)$$

Let $h$ denote the scaling dimension of $\mathcal{O}$. By (C.4), the three-point function has the following expansion:

$$
\begin{aligned}
\langle \mathcal{O}_1(\tau_1) \mathcal{O}_2(\tau_2) \mathcal{O}(L) \rangle &\equiv \frac{C_{12\mathcal{O}}}{|\tau_{12}|^{h_1+h_2-h} (L-\tau_1)^{h+h_{12}} (L-\tau_2)^{h-h_{12}}} \\
&= C_{12\mathcal{O}} \sum_{n=0}^{\infty} \frac{L^{-2h-n}}{|\tau_{12}|^{h_1+h_2-h}} \sum_{k+l=n} \frac{(h+h_{12})_k (h-h_{12})_l \, \tau_1^k \tau_2^l}{k!\, l!}.
\end{aligned}
\tag{C.13}
$$

On the other hand, by (C.3) and (C.11), the three-point function takes on the following form:

$$
\begin{aligned}
\langle \mathcal{O}_1(\tau_1) \mathcal{O}_2(\tau_2) \mathcal{O}(L) \rangle &= C_{12\mathcal{O}} \sum_{n=0}^{\infty} B_n(\tau_1, \tau_2) \langle \partial^n \mathcal{O}(0) \mathcal{O}(L) \rangle \\
&= C_{12\mathcal{O}} \sum_{n=0}^{\infty} B_n(\tau_1, \tau_2) \frac{(2h)_n}{L^{2h+n}}.
\end{aligned}
\tag{C.14}
$$

Matching the coefficients of (C.13) and (C.14) in terms of the expansion in powers of $1/L$, we arrive at the following expression for the OPE kernel:

$$
B_n(\tau_1, \tau_2) = \frac{1}{(2h)_n |\tau_{12}|^{h_1+h_2-h}} \sum_{k+l=n} \frac{(h+h_{12})_k (h-h_{12})_l \, \tau_1^k \tau_2^l}{k!\, l!}.
\tag{C.15}
$$

Inserting the expressions from (C.11) and (C.15) into the four-point function (C.6) and its conformal block expansion (C.10), we obtain the following series representation for the 1D conformal block:

$$
\begin{aligned}
g_{1234,h} &= \left( \frac{1}{|\tau_{12}|^{h_1+h_2} |\tau_{34}|^{h_3+h_4}} \left| \frac{\tau_{14}}{\tau_{24}} \right|^{h_{21}} \left| \frac{\tau_{14}}{\tau_{13}} \right|^{h_{34}} \right)^{-1} \sum_{n=0}^{\infty} B_n(\tau_1, \tau_2) \langle \partial^n \mathcal{O}(0) \mathcal{O}_3(x_3) \mathcal{O}_4(x_4) \rangle \\
&= |\tau_{12}|^h |\tau_{34}|^{h_3+h_4} \left| \frac{\tau_{14}}{\tau_{24}} \right|^{h_{12}} \left| \frac{\tau_{14}}{\tau_{13}} \right|^{h_{43}} \sum_{n=0}^{\infty} \sum_{k+l=n} \frac{(h+h_{12})_k (h-h_{12})_l \, \tau_1^k \tau_2^l}{k!\, l!\, (2h)_n} \\
&\quad \times \sum_{k'+l'=n} \frac{n!}{k'!\, l'!\, |\tau_{34}|^{h_3+h_4-h}} \left( \partial_\tau^{k'} \frac{1}{|\tau-\tau_3|^{h+h_{34}}} \right) \left( \partial_\tau^{l'} \frac{1}{|\tau-\tau_4|^{h-h_{34}}} \right) \Big|_{\tau=0} \\
&= |\tau_{12}|^h |\tau_{34}|^h \left| \frac{\tau_{14}}{\tau_{24}} \right|^{h_{12}} \left| \frac{\tau_{14}}{\tau_{13}} \right|^{h_{43}} \sum_{n=0}^{\infty} \sum_{k+l=n} \frac{(h+h_{12})_k (h-h_{12})_l \, \tau_1^k \tau_2^l}{k!\, l!\, (2h)_n} \\
&\quad \times \sum_{k'+l'=n} \frac{n!\, \mathrm{sign}(\tau_3)^{k'} \mathrm{sign}(\tau_4)^{l'} (h+h_{34})_{k'} (h-h_{34})_{l'}}{k'!\, l'!\, |\tau_3|^{h+h_{34}+k'} |\tau_4|^{h-h_{34}+l'}}.
\end{aligned}
\tag{C.16}
$$

The above expansion of $g_{1234,h}$ looks complicated. However, we know that it is conformally invariant. Consequently, selecting different yet conformally equivalent four-point configurations

$$
(\tau_1, \tau_2, \tau_3, \tau_4) \text{ and } (\tau_1', \tau_2', \tau_3', \tau_4'), \qquad \tau_i' = \frac{a\tau_i + b}{c\tau_i + d}, \qquad \begin{pmatrix} a & b \\ c & d \end{pmatrix} \in GL(2; \mathbb{R}), \tag{C.17}
$$

leads to the same value of $g_{1234,h}$. By choosing some specific configuration, (C.16) may reduce to a much simpler form. Below we will introduce two such configurations.

The first configuration is

$$
\tau_1 = 0, \qquad \tau_2 = z, \qquad \tau_3 = 1, \qquad \tau_4 = \infty. \tag{C.18}
$$

In this case, the cross-ratio (see (57)) is exactly equal to $z$. By (C.16), we have

$$g_{1234,h}(z) = |z|^h {}_2F_1(h - h_{12}, h + h_{34}; 2h; z). \tag{C.19}$$

The second configuration is

$$\tau_1 = \rho, \qquad \tau_2 = -\rho, \qquad \tau_3 = -1, \qquad \tau_4 = 1. \tag{C.20}$$

In this case, the cross-ratio is given by

$$z = \frac{4\rho}{(1+\rho)^2}. \tag{C.21}$$

Then, (C.16) gives

$$g_{1234,h}\left(\frac{4\rho}{(1+\rho)^2}\right) = |4\rho|^h \left|\frac{1-\rho}{1+\rho}\right|^{h_{12}-h_{34}} \sum_{n=0}^{\infty} \frac{n!}{(2h)_n} a_n(h, h_{12}) a_n(h, -h_{34}) \rho^n, \tag{C.22}$$

where the factor $a_n$ is defined as

$$a_n(h, \delta) := \sum_{k=0}^{n} \frac{(-1)^k (h-\delta)_k (h+\delta)_{n-k}}{k!(n-k)!}. \tag{C.23}$$

Through a comparison of (C.19) and (C.23), an insightful identity emerges for $\rho \in (-1, 1)$:

$$_2F_1\left(h - \delta_1, h - \delta_2; 2h; \frac{4\rho}{(1+\rho)^2}\right) = (1+\rho)^{2h} \left(\frac{1-\rho}{1+\rho}\right)^{\delta_1+\delta_2} \sum_{n=0}^{\infty} \frac{n!}{(2h)_n} a_n(h, \delta_1) a_n(h, \delta_2) \rho^n. \tag{C.24}$$

Now let us argue that the domain of validity of (C.24) can be extended to the open unit disc, i.e. $|\rho| < 1$. This can be seen by moving the $1 \pm \rho$ prefactors to the left:

$$(1+\rho)^{-2h} \left(\frac{1+\rho}{1-\rho}\right)^{\delta_1+\delta_2} {}_2F_1\left(h - \delta_1, h - \delta_2; 2h; \frac{4\rho}{(1+\rho)^2}\right) = \sum_{n=0}^{\infty} \frac{n!}{(2h)_n} a_n(h, \delta_1) a_n(h, \delta_2) \rho^n. \tag{C.25}$$

It is well-known that the hypergeometric function $_2F_1(a, b; c; z)$ is analytic in the domain $z \in \mathbb{C} \backslash [1, \infty)$. Mapping to $\rho$ coordinate via $z = \frac{4\rho}{(1+\rho)^2}$, this domain corresponds to $|\rho| < 1$. Therefore, the l.h.s. of (C.25), as a function of $\rho$, is analytic on the open unit disc. Consequently, it has an absolutely convergent power series expansion in terms of $\rho$, which is exactly the r.h.s. of (C.25). This justifies the validity of (C.24) in the whole open unit disc $|\rho| < 1$.

# D   Dimensional reduction of de Sitter scalar free propagators

In this appendix, we show some details of the dimensional reduction of free propagators in $\mathrm{dS}_{d+1}$ into free propagators in $\mathrm{dS}_2$, which was employed in section 4.4 to prove the positivity of the series expansion in the $\rho$ variable for complementary series propagators in $\mathrm{dS}_{d+1}$.

## D.1   Barnes integral for the inversion formula

Let us start from equation (83):

$$\varrho_\lambda^{\mathcal{P}}(\nu) = \frac{\Gamma(\frac{d}{2} \pm i\lambda)}{(4\pi)^{\frac{d-1}{2}} \Gamma(\pm i\nu)} \int_0^{\infty} dz \, \mathbf{F}\left(\frac{1}{2} + i\nu, \frac{1}{2} - i\nu; 1; -z\right) \mathbf{F}\left(\frac{d}{2} + i\lambda, \frac{d}{2} - i\lambda; \frac{d+1}{2}; -z\right). \tag{D.1}$$

To solve this integral, we first apply the Barnes integral representation to the first hypergeometric function, namely

$$\mathbf{F}\left(\frac{1}{2}+i\nu, \frac{1}{2}-i\nu; 1; -z\right) = \frac{1}{\Gamma(\frac{1}{2}\pm i\nu)} \int_{c+i\mathbb{R}} \frac{ds}{2\pi i} \frac{\Gamma(\frac{1}{2}+s\pm i\nu)\Gamma(-s)}{\Gamma(1+s)} z^s. \quad \text{(D.2)}$$

Here, it's essential to have $-\frac{1}{2} < c < 0$ for the validity of the integral. Substituting (D.2) into (83) results in:

$$\varrho_\lambda^{\mathcal{P}}(\nu) = \frac{\nu \sinh(2\pi\nu)\Gamma(\frac{d}{2}\pm i\lambda)}{2^d \pi^{\frac{d+3}{2}}}$$
$$\times \int_{c+i\mathbb{R}} \frac{ds}{2\pi i} \frac{\Gamma(\frac{1}{2}+s\pm i\nu)\Gamma(-s)}{\Gamma(1+s)} \int_0^\infty dz \, \mathbf{F}\left(\frac{d}{2}+i\lambda, \frac{d}{2}-i\lambda; \frac{d+1}{2}; -z\right) z^s, \quad \text{(D.3)}$$

where the $z$ integral is the Mellin transformation of the hypergeometric function:

$$\int_0^\infty dz \, \mathbf{F}\left(\frac{d}{2}+i\lambda, \frac{d}{2}-i\lambda; \frac{d+1}{2}; -z\right) z^s = \frac{\Gamma(s+1)}{\Gamma(\frac{d}{2}\pm i\lambda)} \frac{\Gamma(\frac{d-2}{2}-s\pm i\lambda)}{\Gamma(\frac{d-1}{2}-s)}. \quad \text{(D.4)}$$

This integral is well-defined when $\text{Re}\left(\frac{d}{2}\pm i\lambda\right) > c+1$. Let's define $\Delta_\lambda = \frac{d}{2}+i\lambda$, and choose $\text{Re}(\Delta_\lambda) \geqslant \text{Re}(\bar{\Delta}_\lambda)$, where $\bar{\Delta}_\lambda = d - \Delta_\lambda$. Then this condition is equivalent to $\text{Re}(\bar{\Delta}_\lambda) > c+1$. It's automatically satisfied for $\Delta_\lambda$ in the principal series when $d \geqslant 2$.[11] For $\Delta_\lambda$ in the complementary series, this condition cannot be met when $0 < \Delta_\lambda < \frac{1}{2}$ or $0 < \bar{\Delta}_\lambda < \frac{1}{2}$, which corresponds to the violation of the $L^2$-condition (81) we discussed earlier. For now, let us focus on the case where $\text{Re}(\Delta_\lambda) > \frac{1}{2}$, and therefore, the contour choice is $-\frac{1}{2} < c < \text{Re}(\bar{\Delta}_\lambda) - 1$. By combining (83) and (D.4), we obtain:

$$\varrho_\lambda^{\mathcal{P}}(\nu) = \frac{\nu \sinh(2\pi\nu)}{2^d \pi^{\frac{d+3}{2}}} \int_{c+i\mathbb{R}} \frac{ds}{2\pi i} \frac{\Gamma(\frac{1}{2}+s\pm i\nu)\Gamma(\frac{d-2}{2}-s\pm i\lambda)\Gamma(-s)}{\Gamma(\frac{d-1}{2}-s)}. \quad \text{(D.5)}$$

We want to go further and find an explicit expression for $\varrho_\lambda^{\mathcal{P}}(\nu)$. We will achieve this by closing the contour of integration on the right half of the complex $s$ plane. First of all, notice that by Stirling's approximation, for large real $s$ we get that the integrand goes like $s^{-\frac{5-d}{2}}$. We thus can drop the arc at infinity when closing the contour to the right only if $d < 3$. We will start with that assumption and eventually see that the final answer can be safely analytically continued in $d$. Summing over all residues, we obtain

$$\varrho_\lambda^{\mathcal{P}}(\nu) = \frac{\nu\Gamma(\frac{1}{2}-i\nu)\Gamma(\frac{d-1}{2}-i\nu\pm i\lambda)}{2^d i \pi^{\frac{d+1}{2}}}$$
$$\times \left[ {}_3F_2\left(\begin{matrix} \frac{1}{2}-i\nu, \bar{\Delta}_\lambda-\frac{1}{2}-i\nu, & \Delta_\lambda-\frac{1}{2}-i\nu \\ \frac{d}{2}-i\nu, & 1-2i\nu \end{matrix}; 1\right) - (\nu \to -\nu) \right]. \quad \text{(D.6)}$$

The two ${}_3F_2$ functions that appear in this difference are each divergent as $d \geq 3$, as predicted by the asymptotic behavior of the Mellin integrand. Fortunately, their difference is not divergent. This can be seen by setting the last argument of the hypergeometric functions to be $1-\epsilon$ and performing a series expansion around $\epsilon = 0$. There are no poles in $\epsilon$, and the expression simplifies to:

---

[11]When $d = 1$, it is clear that the $z$ integral in (D.2) gives delta function $\delta(\nu \pm \lambda)$, if $\lambda$ is real.

$$\varrho_\lambda^{\mathcal{P}}(\nu) = \frac{\nu \sinh \pi \nu}{8\pi^{\frac{d+3}{2}}\Gamma\left(\frac{d-1}{2}\right)} \prod_{\pm,\pm} \Gamma\left(\frac{d-1}{4} \pm i\frac{\nu}{2} \pm i\frac{\lambda}{2}\right). \tag{D.7}$$

*Remark* D.1. The expression (D.7) is consistent with condition (81). When $\mathrm{Im}(\lambda) = \pm\frac{d-1}{2}$, the spectral density $\varrho_\lambda^{\mathcal{P}}(\nu)$ has poles in the real axis of $\nu$. This indicates a violation of the square-integrable condition.

While the Källén-Lehmann decomposition formula (82) holds under the conditions $|\mathrm{Im}(\lambda)| < \frac{d-1}{2}$ and $\sigma \in (-\infty, -1]$, its convergence has not been justified for the whole range of $\sigma$ that we are interested in: $\sigma \in \mathbb{C}\backslash[1, +\infty)$. In the next step, we will justify it using the explicit form of the spectral density, as given in eq. (D.7).

Let us consider the same range of $\lambda$ as given in (81), but with $\sigma \in \mathbb{C}\backslash[1, +\infty)$. In this regime of $\sigma$, we can establish the following bound for the integral (82):

$$\left|\int_{\mathbb{R}} d\nu \, \varrho_\lambda^{\mathcal{P}}(\nu) G_\nu^{(1)}(\sigma)\right| \leqslant \int_{\mathbb{R}} d\nu \, \left|\varrho_\lambda^{\mathcal{P}}(\nu)\right| G_\nu^{(1)}(\sigma_*), \tag{D.8}$$

where $\sigma_* \in [-1, 1)$ is defined by

$$\frac{1+\sigma}{2} = \frac{4\rho}{(1+\rho)^2}, \qquad \frac{1+\sigma_*}{2} = \frac{4|\rho|}{(1+|\rho|)^2}. \tag{D.9}$$

This bound follows from the principal-series case in $d = 1$ of proposition 3.1, which we have already proven in section 4.2.

Therefore, it suffices to show the convergence of (82) for $\sigma \in [-1, 1)$. In this regime, we use the following upper bound for $\varrho_\lambda^{\mathcal{P}}(\nu)$ and $G_\nu^{(1)}(\sigma)$:

$$\begin{aligned}
\left|\varrho_\lambda^{\mathcal{P}}(\nu)\right| &\leq A_\lambda (1+|\nu|)^{d-2}, \\
\left|G_\nu^{(1)}(\sigma)\right| &\leq B e^{-\left(2-\sqrt{2(1+\sigma)}\right)\nu} + \frac{C}{1-\sigma}\left(\frac{1+\sigma}{2}\right)^\nu,
\end{aligned} \tag{D.10}$$

where the constants $B$ and $C$ are finite, and $A_\lambda$ is finite for $\lambda$ is in the regime (81). By (D.10), for fixed $\sigma \in [-1, 1)$ and $\lambda$ in the aforementioned regime, the integrand of (82) decays exponentially fast as $\nu$ goes to $\pm\infty$, ensuring the convergence of the integral. Furthermore, by (D.8) and (D.10), the convergence of (82) holds uniformly in a small complex neighborhood of any fixed $\sigma \in \mathbb{C}\backslash[1, +\infty)$, as long as the closure of this small neighborhood does not intersect with the interval $[1, +\infty)$. Consequently, the integral (82) defines an analytic function of $\sigma$ on the single-cut plane $\mathbb{C}\backslash[1, +\infty)$. This justifies the validity of (82) in the regime

$$|\mathrm{Im}(\lambda)| < \frac{d-1}{2}, \quad \sigma \in \mathbb{C}\backslash[1, +\infty). \tag{D.11}$$

## D.2 Analytic continuation in $\lambda$

In this subsection, we elaborate on some details from section 4.4.2 and explain in a more rigorous way to derive equation (87).

To start, let us set a specific positive value, denoted as $a$, and focus on the analytic continuation of (82) from the domain

$$\mathrm{Re}(\lambda) \in (-a, a), \qquad \mathrm{Im}(\lambda) \in \left(\frac{d-1}{2} - \varepsilon, \frac{d-1}{2}\right), \tag{D.12}$$

where $\varepsilon$ is a very small positive number (say $\varepsilon = 0.1$). According to the previous subsection, the well-definedness and analyticity of (82) are already established in this domain. To perform the

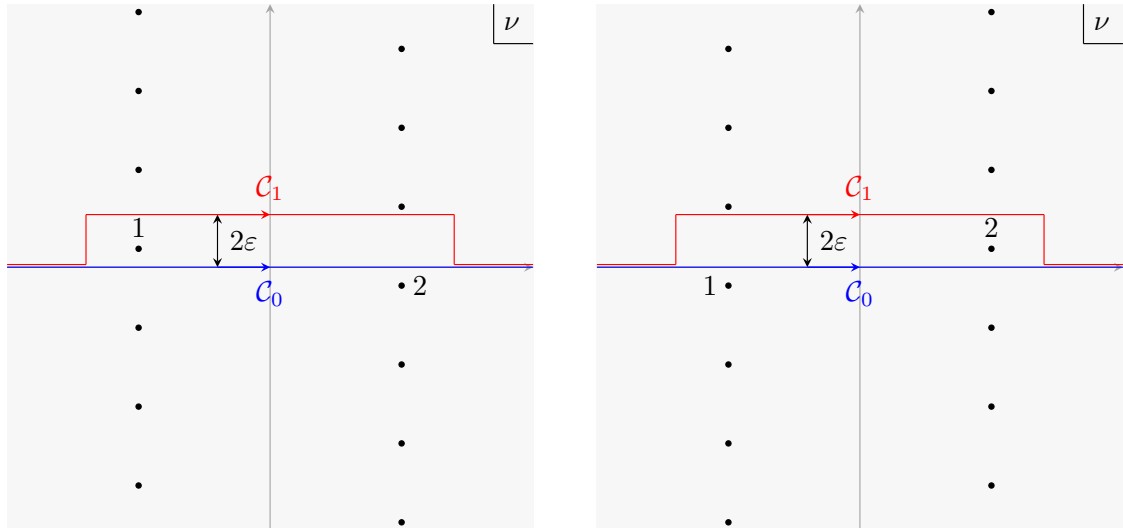

(a) Poles of $\rho_\lambda^{\mathcal{P}}(\nu)$ when $\text{Im}(\lambda) \in (\frac{d-1}{2} - \varepsilon, \frac{d-1}{2})$.    (b) Poles of $\rho_\lambda^{\mathcal{P}}(\nu)$ when $\text{Im}(\lambda) \in (\frac{d-1}{2}, \frac{d-1}{2} + \varepsilon)$.

Figure 8: The pole structure of the spectral density $\rho_\lambda^{\mathcal{P}}(\nu)$. Depending on the imaginary part of $\lambda$, the pole in between contours $\mathcal{C}_0$ and $\mathcal{C}_1$ is pole 1 or pole 2. The contour $\mathcal{C}_0$ runs over the real axis, and corresponds to the integral over the principal series in the Källén-Lehmann representation.

analytic continuation from domain (D.12), we use two crucial observations: (a) the spectral density $\varrho_\lambda^{\mathcal{P}}(\nu)$, as given in (D.7), is a meromorphic function in both $\nu$ and $\lambda$, and (b) the free propagator $G_\nu^{(1)}(\sigma)$ is a meromorphic function in $\nu$. Based on these observations, we have the freedom to deform the integral contour of (82) from the real axis to a specific path composed of piecewise straight lines in the complex plane (see figure 8 for a graphical representation):

$$\mathcal{C}_0: \quad -\infty \rightarrow +\infty \qquad \text{(before)},$$

$$\mathcal{C}_1: \quad -\infty \rightarrow -a-1 \rightarrow -a-1+i\frac{d-1}{2}+2i\varepsilon \tag{D.13}$$

$$\rightarrow a+1+i\frac{d-1}{2}+2i\varepsilon \rightarrow a+1 \rightarrow +\infty \qquad \text{(after)}.$$

By (D.7), the spectral density $\varrho_\lambda^{\mathcal{P}}(\nu)$ has four sets of poles located at

$$\nu = \pm\lambda \pm i\left(\frac{d-1}{2}+2n\right) \qquad (n=0,1,2,\dots), \tag{D.14}$$

corresponding to the four Gamma functions in (D.7). The differences between the contour integrals along $\mathcal{C}_0$ and $\mathcal{C}_1$ is determined by the residue at the pole $\nu = -\lambda + i\frac{d-1}{2}$:[12]

$$\int_{\mathcal{C}_0=\mathbb{R}} d\nu \, \varrho_\lambda^{\mathcal{P}}(\nu)G_\nu^{(1)}(\sigma) = \int_{\mathcal{C}_1} d\nu \, \varrho_\lambda^{\mathcal{P}}(\nu)G_\nu^{(1)}(\sigma) + 2\pi i \operatorname*{Res}_{\nu=-\lambda+i\frac{d-1}{2}} \left[\varrho_\lambda^{\mathcal{P}}(\nu)G_\nu^{(1)}(\sigma)\right]$$

$$= \int_{\mathcal{C}_1} d\nu \, \varrho_\lambda^{\mathcal{P}}(\nu)G_\nu^{(1)}(\sigma) + \frac{\Gamma(-i\lambda)}{2\pi^{\frac{d-1}{2}}\Gamma\left(-\frac{d-1}{2}-i\lambda\right)} G_{-\lambda+i\frac{d-1}{2}}^{(1)}(\sigma). \tag{D.15}$$

Both terms on the r.h.s. of (D.15) are analytical functions of $\lambda$ within the domain:

$$\text{Re}(\lambda) \in (-a,a), \qquad \text{Im}(\lambda) \in \left(\frac{d-1}{2} - \varepsilon, \frac{d-1}{2} + \varepsilon\right). \tag{D.16}$$

---

[12]In this case, the poles from $G_\nu^{(1)}$ do not contribute.

Therefore, by the uniqueness of analytic continuation, we get:

$$
G^{(d)}_\lambda(\sigma) = \int_{\mathcal{C}_1} d\nu \, \varrho^{\mathcal{P}}_\lambda(\nu) G^{(1)}_\nu(\sigma) + \frac{\Gamma(-i\lambda)}{2\pi^{\frac{d-1}{2}}\Gamma\left(-\frac{d-1}{2}-i\lambda\right)} G^{(1)}_{-\lambda+i\frac{d-1}{2}}(\sigma),
$$
$$
\mathrm{Re}(\lambda) \in (-a,a), \quad \mathrm{Im}(\lambda) \in \left(-\frac{d-1}{2}-\varepsilon, \frac{d-1}{2}+\varepsilon\right), \quad \sigma \in \mathbb{C}\backslash[1,+\infty).
$$
(D.17)

Now, let us narrow our focus to the domain:

$$
\mathrm{Re}(\lambda) \in (-a,a), \qquad \mathrm{Im}(\lambda) \in \left(-\frac{d-1}{2}, \frac{d-1}{2}+\varepsilon\right).
$$
(D.18)

In domain (D.18), we deform the integral contour from $\mathcal{C}_1$ back to $\mathcal{C}_0$. The difference between the integrals along these two contours is determined by another pole at $\nu = \lambda - i\frac{d-1}{2}$:

$$
\int_{\mathcal{C}_1=\mathbb{R}} d\nu \, \varrho^{\mathcal{P}}_\lambda(\nu) G^{(1)}_\nu(\sigma) = \int_{\mathcal{C}_0} d\nu \, \varrho^{\mathcal{P}}_\lambda(\nu) G^{(1)}_\nu(\sigma) - 2\pi i \, \mathrm{Res}_{\nu=\lambda-i\frac{d-1}{2}} \left[\varrho^{\mathcal{P}}_\lambda(\nu) G^{(1)}_\nu(\sigma)\right]
$$
$$
= \int_{\mathcal{C}_0} d\nu \, \varrho^{\mathcal{P}}_\lambda(\nu) G^{(1)}_\nu(\sigma) + \frac{\Gamma(-i\lambda)}{2\pi^{\frac{d-1}{2}}\Gamma\left(-\frac{d-1}{2}-i\lambda\right)} G^{(1)}_{\lambda-i\frac{d-1}{2}}(\sigma).
$$
(D.19)

By (D.17) and (D.19), we can express $G^{(d)}_\lambda(\sigma)$ as follows:

$$
G^{(d)}_\lambda(\sigma) = \int_{\mathbb{R}} d\nu \, \varrho^{\mathcal{P}}_\lambda(\nu) G^{(1)}_\nu(\sigma) + \frac{\Gamma(-i\lambda)}{\pi^{\frac{d-1}{2}}\Gamma\left(-\frac{d-1}{2}-i\lambda\right)} G^{(1)}_{-\lambda+i\frac{d-1}{2}}(\sigma),
$$
$$
\mathrm{Re}(\lambda) \in (-a,a), \quad \mathrm{Im}(\lambda) \in \left(\frac{d-1}{2}, \frac{d-1}{2}+\varepsilon\right), \quad \sigma \in \mathbb{C}\backslash[1,+\infty).
$$
(D.20)

Here we have used the fact that $G^{(1)}_\lambda = G^{(1)}_{-\lambda}$.

The r.h.s. of (D.20) is precisely the form we desire. To extend the domain of $\lambda$ for which (D.20) is valid, we exploit the fact that when $\nu$ is real, the spectral density $\varrho^{\mathcal{P}}_\lambda(\nu)$ remains analytic in $\lambda$ as long as $\frac{d-1}{2} < \mathrm{Im}(\lambda) < \frac{d}{2}$. Additionally, the second term on the r.h.s. of (D.20) is also analytic in $\lambda$ within the same range. Thus, we conclude that

$$
G^{(d)}_\lambda(\sigma) = \int_{\mathbb{R}} d\nu \, \varrho^{\mathcal{P}}_\lambda(\nu) G^{(1)}_\nu(\sigma) + \frac{\Gamma(-i\lambda)}{\pi^{\frac{d-1}{2}}\Gamma\left(-\frac{d-1}{2}-i\lambda\right)} G^{(1)}_{-\lambda+i\frac{d-1}{2}}(\sigma),
$$
$$
\mathrm{Im}(\lambda) \in \left(\frac{d-1}{2}, \frac{d}{2}\right), \quad \sigma \in \mathbb{C}\backslash[1,+\infty).
$$
(D.21)

A similar result can be obtained for the other domain of $\lambda$, using either a similar argument or the symmetry $G^{(d)}_\lambda = G^{(d)}_{-\lambda}$. The expression is as follows:

$$
G^{(d)}_\lambda(\sigma) = \int_{\mathbb{R}} d\nu \, \varrho^{\mathcal{P}}_\lambda(\nu) G^{(1)}_\nu(\sigma) + \frac{\Gamma(i\lambda)}{\pi^{\frac{d-1}{2}}\Gamma\left(-\frac{d-1}{2}+i\lambda\right)} G^{(1)}_{\lambda+i\frac{d-1}{2}}(\sigma),
$$
$$
\mathrm{Im}(\lambda) \in \left(-\frac{d}{2}, -\frac{d-1}{2}\right), \quad \sigma \in \mathbb{C}\backslash[1,+\infty).
$$
(D.22)

### D.3 Flat space limit

In flat space, the spectral density corresponding to the dimensional reduction from $\mathbb{R}^{d,1}$ to $\mathbb{R}^{1,1}$ follows trivially from the momentum space representation. Denote the Green function of a free scalar with mass $M$ in $\mathbb{R}^{d,1}$ by $G_M^{(d)}(x)$. It can be expressed as the following Fourier transformation

$$G_M^{(d)}(|x|) = \int \frac{d^{d+1}p}{(2\pi)^{d+1}} \frac{1}{p^2 + M^2} e^{ip\cdot x}. \tag{D.23}$$

Consider the special case with $x^\mu = (\hat{x}, 0, \dots, 0)$, where $\hat{x}$ is a vector in $\mathbb{R}^{1,1}$. Then it is natural to take $p^\mu = (\hat{p}, \vec{k})$ with $\vec{k} \in \mathbb{R}^{d-1}$, and hence $G_M^{(d)}(x)$ can be rewritten as

$$G_M^{(d)}(|x|) = \int \frac{d^{d-1}\vec{k}}{(2\pi)^{d-1}} \int \frac{d^2\hat{p}}{(2\pi)^2} \frac{1}{\hat{p}^2 + M^2 + \vec{k}^2} e^{i\hat{p}\cdot\hat{x}}. \tag{D.24}$$

Defining $m = \sqrt{M^2 + \vec{k}^2}$ and treating it as a mass, we can identify the $\hat{p}$ integral as the Green function $G_m^{(1)}$ in $\mathbb{R}^{1,1}$. The remaining integral over $\vec{k}$ then becomes an integral over $m$ multiplied by volume of $S^{d-2}$. Altogether, we have

$$G_M^{(d)}(|x|) = \int_0^\infty dm^2 \varrho_M^{\mathbb{M}}(m^2) G_m^{(1)}(|x|), \qquad \varrho_M^{\mathbb{M}}(m^2) = \Theta(m^2 - M^2) \frac{(m^2 - M^2)^{\frac{d-3}{2}}}{(4\pi)^{\frac{d-1}{2}} \Gamma(\frac{d-1}{2})}, \tag{D.25}$$

where $\Theta$ denotes the step function.

Next, we are going to show that eq. (D.25) can be recovered by taking the flat space limit of (D.7). We follow the procedure described in [17,65] by restoring the factors of the de Sitter radius $R$ and taking $\lambda \sim RM$ and $\nu \sim Rm$

$$\varrho_M^{\mathbb{M}}(m^2) = \lim_{R\to\infty} \frac{R}{m} \varrho_{RM}^{\mathcal{P}}(Rm), \tag{D.26}$$

where to restore the correct factors of the radius we need to take

$$\varrho_\lambda^{\mathcal{P}}(\nu) \to R^{-(d-1)} \varrho_\lambda^{\mathcal{P}}(\nu). \tag{D.27}$$

We thus have

$$\varrho_M^{\mathbb{M}}(m^2) = \lim_{R\to\infty} \frac{R^{3-d}\sinh(\pi mR)}{8\pi^{\frac{d+3}{2}}\Gamma(\frac{d-1}{2})} \prod_{\pm,\pm} \Gamma\left(\frac{d-1}{4} \pm i\frac{m}{2}R \pm i\frac{M}{2}R\right). \tag{D.28}$$

To evaluate this limit, we use

$$\Gamma(a + iR)\Gamma(a - iR) \underset{R\to\infty}{\sim} 2\pi e^{-\pi R} R^{2a-1}, \tag{D.29}$$

and we obtain

$$\begin{aligned}
\varrho_M^{\mathbb{M}}(m^2) &= \frac{(m^2 - M^2)^{\frac{d-3}{2}}}{(4\pi)^{\frac{d-1}{2}}\Gamma(\frac{d-1}{2})} \lim_{R\to\infty} e^{\pi mR} e^{-\frac{\pi}{2}[(m+M)+|m-M|]R} \\
&= \frac{(m^2 - M^2)^{\frac{d-3}{2}}}{(4\pi)^{\frac{d-1}{2}}\Gamma(\frac{d-1}{2})} \Theta(m^2 - M^2),
\end{aligned} \tag{D.30}$$

reproducing (D.25).

# E A group theoretical analysis of the dimensional reduction from $SO(d+1,1)$ to $SO(d,1)$

In section 4, by directly expanding a $(d+1)$ dimensional Green function into 2D Green functions, we find that discrete series does not contribute to this dimensional reduction, which is consistent with the conjecture made in appendix A. In this appendix, we provide a purely group theoretical explanation of this fact. More precisely, we will prove the following proposition:

> **Proposition E.1.** *Given a scalar principal or complementary series representation R of $SO(d+1,1)$, the only allowed UIRs of $SO(d,1)$ in the restricted representation $R|_{SO(d,1)}$ are scalar principal and complementary series.*

## E.1 A quick review of $SO(d+1,1)$ (scalar) UIRs

We choose $SO(d+1,1)$ generators to be $L_{AB} = -L_{BA}, 0 \leqslant A, B \leqslant d+1$ satisfying commutation relations

$$[L_{AB}, L_{CD}] = \eta_{BC} L_{AD} - \eta_{AC} L_{BD} + \eta_{AD} L_{BC} - \eta_{BD} L_{AC}, \tag{E.1}$$

where $\eta_{AB}$ is given by eq. (2). In a unitary representation, $L_{AB}$ are realized as anti-hermitian operators on some Hilbert space. The isomorphism between $\mathfrak{so}(d+1,1)$ and the $d$-dimensional Euclidean conformal algebra is realized as

$$L_{ij} = M_{ij}, \qquad L_{0,d+1} = D, \qquad L_{d+1,i} = \frac{1}{2}(P_i + K_i), \qquad L_{0,i} = \frac{1}{2}(P_i - K_i). \tag{E.2}$$

The commutation relations of the conformal algebra following from (E.1) and (E.2) are

$$
\begin{aligned}
[D, P_i] &= P_i, & [D, K_i] &= -K_i, & [K_i, P_j] &= 2\delta_{ij} D - 2M_{ij}, \\
[M_{ij}, P_k] &= \delta_{jk} P_i - \delta_{ik} P_j, & [M_{ij}, K_k] &= \delta_{jk} K_i - \delta_{ik} K_j, \\
[M_{ij}, M_{k\ell}] &= \delta_{jk} M_{i\ell} - \delta_{ik} M_{j\ell} + \delta_{i\ell} M_{jk} - \delta_{j\ell} M_{ik}.
\end{aligned}
\tag{E.3}
$$

The quadratic Casimir of $SO(d+1,1)$, which commutes with all $L_{AB}$, is chosen to be

$$C^{SO(d+1,1)} = \frac{1}{2} L_{AB} L^{AB} = D(d-D) + P_i K_i + C^{SO(d)}. \tag{E.4}$$

Here $C^{SO(d)} \equiv \frac{1}{2} M_{ij} M^{ij}$ is the quadratic Casimir of $SO(d)$ and it is negative-definite for a unitary representation since $M_{ij}$ are anti-hermitian. For example, for a spin-$s$ representation of $SO(d)$, it takes the value of $-s(s+d-2)$.

Next we describe the representation $\mathcal{F}_\Delta$ in detail, which amounts to specifying the representation space, the action, and the inner product. First, as a vector space, $\mathcal{F}_\Delta$ consists of smooth wavefunctions $\psi(x)$ on $\mathbb{R}^d$, that decay as $\mathcal{O}\left(|x|^{-2\Delta}\right)$ at $\infty$. Second, the action of $SO(d+1,1)$ generators on $\psi(x)$ is the same as in a conformal field theory

$$
\begin{aligned}
P_i \psi(x) &= -\partial_i \psi(x), & K_i \psi(x) &= \left(x^2 \partial_i - x_i(x \cdot \partial_x + \Delta)\right) \psi(x), \\
D\psi(x) &= -(x \cdot \partial_x + \Delta)\psi(x), & M_{ij}\psi(x) &= \left(x_i \partial_j - x_j \partial_i\right)\psi(x).
\end{aligned}
\tag{E.5}
$$

From eq. (E.5), we can find that $C^{SO(d+1,1)}$ takes the value $\Delta(d-\Delta)$ in $\mathcal{F}_\Delta$. The $SO(d+1,1)$ invariant inner product on $\mathcal{F}_\Delta$ is uniquely fixed (up to an overall normalization) by requiring this action to be anti-hermitian. In particular, when $\Delta \in \frac{d}{2} + i\mathbb{R}$, the inner product is nothing but the standard $L_2$ inner product on $\mathbb{R}^d$, and when $\Delta \in (0, d)$, the inner product becomes

$$(\psi_1, \psi_2)_{C_\Delta} = \int d^d x_1 d^d x_2 \frac{\psi_1^*(x_1)\psi_2(x_2)}{|x_1 - x_2|^{2\bar{\Delta}}}, \qquad \bar{\Delta} = d - \Delta. \tag{E.6}$$

The restricted representation of $\mathcal{F}_\Delta$ to the maximal compact subgroup $SO(d+1)$ of $SO(d+1,1)$ is given by $\bigoplus_{n\in\mathbb{N}}\mathbb{Y}_n$, where $\mathbb{Y}_n$ is the spin $n$ representation of $SO(d+1)$.[13] The scalar UIRs of $SO(d+1,1)$ are characterized by the property that their $SO(d+1)$ content only contains single-row Young tableau. Apart from principal and complementary series, there is another class of scalar UIRs, which is called type I exceptional series in [49] and is denoted by $\mathcal{V}_{p,0}$ with $p$ being a positive integer. Roughly speaking, $\mathcal{V}_{p,0}$ can be realized as an $\infty$ dimensional invariant subspace of $\mathcal{C}_{d+p-1}$. The inner product is (E.6) is positive definite when restricted to $\mathcal{V}_{p,0}$ but not in the larger space $\mathcal{C}_{d+p-1}$. The $SO(d+1)$ content of $\mathcal{V}_{p,0}$ consists of $\mathbb{Y}_p, \mathbb{Y}_{p+1}, \mathbb{Y}_{p+2},\ldots$. A more detailed and precise construction of the type $\mathcal{V}$ exceptional series can be found in [48] (where it is denoted by $F_{0,\nu}$ with $\nu$ being the same as $p$ here) and [49]. When $d=1$, $\mathcal{V}_{p,0}$ becomes reducible, i.e. $\mathcal{V}_{p,0}=\mathcal{D}_p^+\oplus\mathcal{D}_p^+$, where $\mathcal{D}_p^\pm$ are the highest/lowest-weight discrete series representations of $SO(2,1)$.

## E.2 Details of the proof

Consider a (scalar) principal or complementary series representation $\mathcal{F}_\Delta$. First, we know that the $SO(d+1)$ components of $\mathcal{F}_\Delta$ are all $\mathbb{Y}_n$. Because of the branching rule from $SO(d+1)$ to $SO(d)$, it is clear that the $SO(d)$ components of $\mathcal{F}_\Delta$ are also the single-row Young tableaux. Therefore the restriction $\mathcal{F}_\Delta|_{SO(d,1)}$ cannot contain anything beyond the scalar UIRs of $SO(d,1)$. Our next step is to prove the absence of the type $\mathcal{V}$ exceptional series of $SO(d,1)$ in this restriction.[14]

To sketch the main idea of the proof, it would be convenient to switch to the ket notation. For a $\mathcal{V}_{\ell,0}$ of $SO(d,1)$, there exists some nonvanishing state $|\psi\rangle_{i_1\cdots i_\ell}\in\mathcal{V}_{\ell,0}$ that carries the spin $\ell$ representation of $SO(d)$. In other words, the indices $(i_1\cdots i_\ell)$ are symmetric and traceless. Acting $L_{0,i_1}$ on this state and summing over $i_1$ from 1 to $d$, we obtain a state that has the spin $(\ell-1)$ symmetry. On the other hand, as reviewed above, $\mathcal{V}_{\ell,0}$ of $SO(d,1)$ does not contain the spin $(\ell-1)$ representation of $SO(d)$. So this state must vanish, i.e. $L_{0,i_1}|\psi\rangle_{i_1\cdots i_\ell}=0$. In the remaining part of this section, we will show that given any $|\psi\rangle_{i_1\cdots i_\ell}$ in $\mathcal{F}_\Delta$ that carries the spin $\ell$ representation of $SO(d)$, imposing $L_{0,i_1}|\psi\rangle_{i_1\cdots i_\ell}=0$ leads to the vanishing of $|\psi\rangle_{i_1\cdots i_\ell}$ itself. This property contradicts the existence of any type $\mathcal{V}$ exceptional series in $\mathcal{F}_\Delta$.

Now let's switch back to the wavefunction picture. Then the analogue of $|\psi\rangle_{i_1\cdots i_\ell}$ should be a wavefunction $\psi(x)$ that transforms as a spin $\ell$ tensor under $SO(d)$. The spin $\ell$ condition can be easily imposed by introducing a null vector $z^i\in\mathbb{C}^d$:

$$\psi(x)=g(r)(x\cdot z)^\ell, \qquad r=\sqrt{x^2}. \tag{E.7}$$

There is a basis of such wavefunctions, labelled by an integer $n\geqslant\ell$. The reason is that every $\mathbb{Y}_n$ of $\mathcal{F}_\Delta$ contains exactly one copy of the spin $\ell$ representation of $SO(d)$ when $n\geqslant\ell$. Denote the basis by $\psi_{n\ell}(x)=g_{n\ell}(r)(x\cdot z)^\ell$, and each $\psi_{n\ell}(x)$ should satisfy the Casimir equation

$$C^{SO(d+1)}\psi_{n\ell}(x)=-n(n+d-1)\psi_{n\ell}(x). \tag{E.8}$$

By construction, the $SO(d+1)$ Casimir is

$$C^{SO(d+1)}=C^{SO(d)}+L_{i,d+1}^2=C^{SO(d)}+\frac{1}{4}(P_i+K_i)^2, \tag{E.9}$$

where $C^{SO(d)}=-\ell(\ell+d-2)$ when acting on $\psi_{n,\ell}$, and the explicit form of $P_i+K_i$ follows from eq. (E.5)

$$P_i+K_i=(r^2-1)\partial_i-2x_i(x\cdot\partial_x+\Delta). \tag{E.10}$$

---

[13] When $d=1$, $\mathbb{Y}_n$ should be understood as the direct sum of the spin $\pm n$ representations of $SO(2)$ for any $n\geqslant 1$.

[14] When $d=1$, the argument below can be used to exclude discrete series.

By solving eq. (E.8) we obtain

$$g_{n,\ell}(r) = \frac{1}{(1+r^2)^{\Delta+n}} \, {}_2F_1\left(\ell-n, 1-n-\frac{d}{2}; \frac{d}{2}+\ell; -r^2\right), \tag{E.11}$$

where the hypergeometric function is a monic polynomial of $-r^2$ of degree $n-\ell$. In particular, $g_{\ell,\ell}(r) = \frac{1}{(1+r^2)^{\Delta+\ell}}$. Altogether, the most general $\psi(x) \in \mathcal{F}_\Delta$ that furnishes the $\ell$ representation of $SO(d)$ should take the form

$$\psi(x) = \sum_{n \geqslant \ell} c_n \, \psi_{n,\ell}(x). \tag{E.12}$$

In the wavefunction picture, the condition $L_{0i_1}|\psi\rangle_{i_1\cdots i_\ell} = 0$ becomes

$$L_{0i}\mathcal{D}_i \psi(x) = 0, \qquad \mathcal{D}_i = \partial_{z^i} - \frac{1}{d + 2(z\cdot\partial_z - 1)} z_i \, \partial_z^2, \tag{E.13}$$

where $\mathcal{D}_i$ is the interior derivative, used to strip off $z^i$ while respecting its nullness [48], and the differential operator realization of $L_{0i}$ can be derived from eq. (E.5)

$$L_{0i} = \frac{1}{2}(P_i - K_i) = -\frac{1+r^2}{2}\partial_i + x_i(x\cdot\partial_x + \Delta). \tag{E.14}$$

The most important step in our proof is computing $L_{0i}\mathcal{D}_i\psi_{n,\ell}(x)$. Before starting doing any real calculation, we recall that acting with any $L_{0a}$ on a state in $\mathbb{Y}_n$ yields another state belonging to $\mathbb{Y}_{n-1} \oplus \mathbb{Y}_{n+1}$, which was shown in [49, 63]. Using this fact, we can easily conclude that $L_{0i}\mathcal{D}_i\psi_{n,\ell}(x)$ is a linear combination of $\psi_{n-1,\ell-1}(x)$ and $\psi_{n+1,\ell-1}(x)$, i.e.

$$L_{0i}\mathcal{D}_i\psi_{n,\ell}(x) = \alpha_{n,\ell}\psi_{n+1,\ell-1}(x) + \beta_{n,\ell}\psi_{n-1,\ell-1}(x), \tag{E.15}$$

where $\alpha_{n,\ell}$ and $\beta_{n,\ell}$ are constants to be determined. For the l.h.s, we first compute the action of $\mathcal{D}_i$

$$\mathcal{D}_i\psi_{n,\ell}(x) = \ell \, g_{n,\ell}(r)\left[x_i(x\cdot z)^{\ell-1} - \frac{\ell-1}{d+2(\ell-2)}z_i x^2(x\cdot z)^{\ell-2}\right], \tag{E.16}$$

and then plugging in (E.14) yields

$$L_{0i}\mathcal{D}_i\psi_{n,\ell}(x) = \frac{\ell(d+\ell-3)}{d+2(\ell-2)}\mathfrak{D}_\ell \, g_{n,\ell}(y)(x\cdot z)^{\ell-1}, \tag{E.17}$$

where we have made the substitution $y = -r^2$, and defined a first-order differential operator $\mathfrak{D}_\ell$ in terms of $y$

$$\mathfrak{D}_\ell = -(1+y)y\partial_y - \left[(\Delta+\ell)y + \left(\frac{d}{2}+\ell-1\right)(1-y)\right]. \tag{E.18}$$

Eq. (E.15) implies that $\mathfrak{D}_\ell \, g_{n,\ell}(y)$ is a linear combination of $g_{n\pm1,\ell-1}$. We can easily fix the combination coefficients simply by studying the behavior of $\mathfrak{D}_\ell$ near $y=0$ and $y=1$. The result is

$$\mathfrak{D}_\ell g_{n,\ell} = -\frac{\frac{d}{2}+\ell-1}{d+2n-1}\left((\Delta+n)g_{n+1,\ell-1} + (\bar\Delta+n-1)g_{n-1,\ell-1}\right), \tag{E.19}$$

where $\bar\Delta = d - \Delta$. This identity can also be checked by using contiguous relations of the hypergeometric function. Altogether, by combining (E.17) and (E.19), we get

$$\alpha_{n,\ell} = -\frac{\ell(d+\ell-3)}{d+2(\ell-2)}\frac{\frac{d}{2}+\ell-1}{d+2n-1}(\Delta+n),$$

$$\beta_{n,\ell} = -\frac{\ell(d+\ell-3)}{d+2(\ell-2)}\frac{\frac{d}{2}+\ell-1}{d+2n-1}(\bar\Delta+n-1). \tag{E.20}$$

Therefore, the condition $L_{0i}\mathcal{D}_i\psi(x) = 0$ yields a recurrence relation

$$c_n\alpha_{n,\ell} + c_{n+1}\beta_{n+1,\ell} = 0, \qquad n \geq \ell, \tag{E.21}$$

together with the initial condition $c_\ell\beta_{\ell,\ell} = 0$. Since the $\alpha$'s and $\beta$'s are nonvanishing, all $c_n$ have to vanish identically, and hence $\psi(x) = 0$.

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
