# Peer review of "A radial variable for de Sitter two-point functions"

_SciPost Physics, doi:SciPost Phys. 18, 164 (2025)_

## Round 1 · Referee Report · Anonymous (Referee 1) · 2024-12-4

Strengths

1 - Develops a new coordinate system to represent the two-point correlators in dS.

2 - these new coordinates allow the correlators to be written in terms of a convergent expansion in the radial variable

3 - the convergence of the series makes proving analytic properties or positivity more straightforward.

4 - Although these results are only applied to the free-propagators, they can be used for any correlator via the Kallen-Lehmann representation

Weaknesses

1 - The demonstration of the utility of this new variable using a concrete example would have been helpful.

Report

The use of the Kallen-Lehmann representation and analytic continuation from the sphere to dS loop calculations has proven to be a powerful tool for understand loop corrections in de Sitter space. This paper introduces a new representation of the free Green's functions that are exponentially convergent, allowing one to prove a number of useful properties of the analytic properties of the two-point statistic in dS. The paper lacks physics examples where this new variable is useful, but given the existing literature with the many worked examples, many readers will know how to apply these techniques. I recommend publication in the current form.

Recommendation

Publish (surpasses expectations and criteria for this Journal; among top 10%)

---

## Round 1 · Referee Report · Anonymous (Referee 2) · 2025-4-29

Strengths

This paper explores technical aspects of QFT correlators in de Sitter spacetime. The main result is a new representation of the 2-point function, taking inspiration from CFT correlators.
I found the discussion to be clear, if somewhat technical.

Weaknesses

I agree with referee 1 that examples of the utility of the new parameterization would have been useful.

Report

I am happy to recommend the paper for publication.

Recommendation

Publish (meets expectations and criteria for this Journal)

---

## Editorial Decision

published